# A Neighborhood Rough Sets-Based Attribute Reduction Method Using Lebesgue and Entropy Measures

**DOI:** 10.3390/e21020138

**Published:** 2019-02-01

**Authors:** Lin Sun, Lanying Wang, Jiucheng Xu, Shiguang Zhang

**Affiliations:** 1College of Computer and Information Engineering, Henan Normal University, Xinxiang 453007, Henan, China; 2School of Computer Science and Technology, Tianjin University, Tianjin 300072, China

**Keywords:** rough sets, neighborhood rough sets, attribute reduction, neighborhood entropy, Lebesgue measure

## Abstract

For continuous numerical data sets, neighborhood rough sets-based attribute reduction is an important step for improving classification performance. However, most of the traditional reduction algorithms can only handle finite sets, and yield low accuracy and high cardinality. In this paper, a novel attribute reduction method using Lebesgue and entropy measures in neighborhood rough sets is proposed, which has the ability of dealing with continuous numerical data whilst maintaining the original classification information. First, Fisher score method is employed to eliminate irrelevant attributes to significantly reduce computation complexity for high-dimensional data sets. Then, Lebesgue measure is introduced into neighborhood rough sets to investigate uncertainty measure. In order to analyze the uncertainty and noisy of neighborhood decision systems well, based on Lebesgue and entropy measures, some neighborhood entropy-based uncertainty measures are presented, and by combining algebra view with information view in neighborhood rough sets, a neighborhood roughness joint entropy is developed in neighborhood decision systems. Moreover, some of their properties are derived and the relationships are established, which help to understand the essence of knowledge and the uncertainty of neighborhood decision systems. Finally, a heuristic attribute reduction algorithm is designed to improve the classification performance of large-scale complex data. The experimental results under an instance and several public data sets show that the proposed method is very effective for selecting the most relevant attributes with high classification accuracy.

## 1. Introduction

Over the past few decades, data classification has become one of the important aspects of data mining, machine learning, pattern recognition, etc. As an important application of rough set models in a variety of practical problems, attribute reduction methods in information systems have been drawing wide attention of researchers [1,2]. It is a fundamental research theme in the field of granular computing [3]. Since a lot of information is gathered and it may include a large number of redundant and noisy attributes, the main objective of attribute reduction based on rough sets is to eliminate the redundant attributes, classify data and extract useful information [4].

Attribute reduction in rough set theory has been recognized as an important feature selection method [2]. Considering whether the evaluation criterion involves classification models, the existing feature selection methods can be broadly classified into the following three categories [5]: filter, wrapper and embedded methods. Based on the intrinsic properties of data set, the filter methods select a feature subset as a preprocessing step that is independent of the learning algorithm [6]. Lyu et al. [7] investigated a filter method based on maximal information coefficient, which eliminates redundant information that does not require additional processes. The wrapper methods use a classifier to find the most discriminant feature subset by minimizing an error prediction function [8]. Jadhav et al. [9] designed a wrapper feature selection method and performed functional ranking based on information gain directed genetic algorithm. Unfortunately, the wrapper methods not only exhibit sensitivity to the classifier, but also tend to consume a lot of runtime [10]. Hence, few works in the field employ these methods. The embedded methods integrate feature selection in the training process to reduce the total time required for reclassifying different subsets [11]. Imani et al. [12] introduced an embedded algorithm based on a Chisquare feature selector. But it is not as accurate as the wrapper in classification problems. On the basis of the above analysis, our attribute reduction method is based on the filter method, in which a heuristic search algorithm is used to find an optimal attribute reduction subset for complex data sets using neighborhood rough set model. 

Rough set theory has become an efficient mathematical tool for attribute reduction to discover data dependencies and reduce redundancy attributes contained in data sets [13,14]. Gu et al. [15] proposed a kernelized fuzzy rough set, but the result is critically depending on setting control parameters and the design of objective function. Raza and Qamar [16] presented a parallel rough sets-based dependency calculation method for feature selection, and it could directly find the positive region-based objects without calculating the positive region itself. However, rough sets can only deal with attributes of a specific type in information systems using a binary relation [17]. In addition, the traditional rough set model need to discretize the data when dealing with continuous data, but the process ignores the differences among data and affects the information expression of the original attribute set to a certain degree [18]. Moreover, the original property of the continuous-valued data will change after discretization, and some useful information will be lost [8]. To overcome this drawback, scholars have developed many extensions for the traditional rough set model [19,20]. As an extended rough set model, neighborhood rough set model is introduced to solve the problem that classical rough sets cannot handle continuous numerical data. Since most of data in attribute reduction are numerical, when utilizing neighborhood rough sets, the discretization of continuous data can be avoided [21]. Hu et al. [22] developed a neighborhood rough set model via the *δ*-neighborhood set to deal with discrete and continuous data sets. Chen et al. [13] proposed a neighborhood rough sets-based feature reduction fish swarm algorithm to deal with numerical data sets. Sun et al. [23] studied a gene selection algorithm based on Fisher linear discriminant and neighborhood rough sets, which is of great practical significance for cancer clinical diagnosis. Mu et al. [24] investigated a gene selection method using Fisher transformation based on neighborhood rough sets for numerical data sets. Li et al. [14] developed a feature reduction method based on neighborhood rough sets and discernibility matrix, but there is a hypothesis that all features data are available. Nonetheless, since the global neighborhood in this field is only used to deal with decision systems, that is, each sample uses the same neighborhood value in different conditional attribute combinations; this method has a high time complexity and does not result in the optimal *δ* value [24]. Wang et al. [1] constructed local neighborhood rough sets to deal with labeled data. It is well known that the neighborhood rough sets can be employed to deal with an information system with heterogeneous attributes including categorical and numerical attributes [22]. However, a large number of existing attribute reduction algorithms based on rough set model and its variations only analyze finite sets, which would limit their application to some extension. Halmos [25] used Lebesgue measure as measure theory to achieve uncertainty measures. Song and Li [26] stated Lebesgue theorem in non-additive measure theory. Xu et al. [27] introduced Lebesgue integral over infinite interval and presented a computation method based on kernel function for uncertainty measures. Halcinová et al. [28] investigated the weighted Lebesgue integral by Lebesgue differentiation theorem, and used Lebesgue measure to develop the standard weighted *L_p_*-based sizes. Park et al. [29] expressed a measurable map through Lebesgue integration to define the Cumulative Distribution Transform of density functions. Recently, scholars [30,31,32] introduced Lebesgue measure as a promising additional method to resolve some problems in the application of data analysis. As inspired by the ideas of Lebesgue measure that can efficiently evaluate infinite sets, it is therefore necessary to employ Lebesgue measure to study uncertainty measures and efficient reduction algorithms for infinite sets in neighborhood decision systems. 

Uncertainty measures play an important role in the uncertainty analysis of granular computing models [33]. Ge et al. [34] researched a positive region-based reduction algorithm from the relative discernibility perspective in rough set theory. Sun and Xu [35] proposed a positive region-based granular space for feature selection based on rough sets. Nevertheless, the positive region in these models only draws attention to the consistent samples whose similarity classes are completely contained in some decision classes [36]. Meng et al. [37] presented an intersection neighborhood for numerical data, and designed a gene selection algorithm using positive region and gene ranking based on neighborhood rough sets. Liu et al. [38] designed a hash-based algorithm to calculate the positive region of neighborhood rough sets for attribute reduction. Li et al. [39] investigated a positive region-based related attribute reduction in neighborhood-based decision-theoretic rough set model. In summary, these literatures about attribute reduction are all based on rough sets or neighborhood rough sets from algebra view. And some of the abovementioned algorithms can achieve optimal reducts with criterion preservation, but in a sense, their reducts still have some redundant attributes that can be further deleted [2]. What’s more, a lot of reduction algorithms still have higher time expenses when dealing with high-volume and high-dimensional data sets.

Until now, the attribute reduction models based on information theory have been studied extensively. Information entropy introduced by Shannon [40] is a very useful method for the representation of information contents in various domains [41]. Liang et al. [42] proposed information entropy, rough entropy and knowledge granulation for classifications in an incomplete information system. Sun et al. [43] presented some rough entropy-based uncertainty measures to improve computational efficiency of a heuristic feature selection algorithm. Gao et al. [2] introduced an attribute reduction algorithm based on the maximum decision entropy in decision-theoretic rough set model. Nevertheless, most of traditional rough sets-based methods for attribute reduction in the information view are not suitable to measure neighborhood classes of a real-value data set [20]. Liu et al. [44] combined information entropy with neighborhood rough sets to develop neighborhood mutual information. Chen et al. [20] constructed a gene selection method based on neighborhood rough sets and neighborhood entropy gain measures. Wang et al. [45] presented a feature selection method based on discrimination measures using conditional discrimination index in neighborhood rough sets. However, it is noted that the monotonicity of the abovementioned uncertainty measures does not always hold. In general, the attribute significance measures constructed in these algorithms can be applicable for numeric data sets, and these literatures for attribute reduction are based on rough sets or neighborhood rough sets from the information view. 

As we can see, many existing methods of attribute reduction in neighborhood rough sets usually only start from the algebraic point of view or the information point of view, while the definition of attribute significance based on algebraic view only describes the effect of attributes on the subset of classification contained [46]. The definition of attribute significance based on information view only describes the influence of attributes on the uncertain classification subset contained in the domain and suitable for small-scale data sets [47]. Thus, they each have certain limitations in the real-world application. To overcome the shortcoming, considering to efficiently combine the above two views, Wang et al. [46] studied rough reduction in both the algebra view and the information view, and illustrated the definition of reduction and relative reduct in both the algebra view and the information view. Attribute reduction algorithms under the algebra and information viewpoints in rough set theory have been enhanced by filtering out redundant objects by Qian et al. [48]. Although these methods have their own advantages, they are still inefficient and not suitable for reducing large-scale high-dimensional data, and the enhanced algorithms only decrease the computation time to a certain extent [47]. Inspired by this, to study neighborhood rough sets from the two views and achieve great uncertainty measures in neighborhood decision systems, the algebra view and the information view will be combined to develop attribute reduction algorithm for infinite sets in continuous-valued data sets. It follows that the Lebesgue measure [25] is introduced into neighborhood entropy to investigate the uncertainty measures in neighborhood decision systems, an attribute reduction method using Lebesgue and entropy measures is presented, and then a heuristic search algorithm is designed to analyze the uncertainty and noisy of continuous and complex data sets.

The rest of this paper is structured as follows: Some related concepts are briefly reviewed in Section 2. Section 3 describes Lebesgue measure-based neighborhood uncertainty measures, neighborhood entropy-based uncertainty measures, and comparison analysis with two representative reducts. Then, an attribute reduction algorithm with complexity analysis is presented. In Section 4, the classification experiments are conducted on public five UCI data sets and four gene expression data sets. Finally, Section 5 summarizes this study.

## 2. Previous Knowledge

In this section, we introduce some basic concepts and properties of decision system. The detailed descriptions can be found in literatures [22,41,49,50].

### 2.1. Rough Sets

Given a decision system *DS* = <*U*, *C*, *D*, *V*, *f*>, usually written more simply as *DS* = <*U*, *C*, *D*>, where *U* = {*x*_1_, *x*_2_, ⋯, *x_l_*} is a sample set named universe; *C* = {*a*_1_, *a*_2_, ⋯, *a**_s_*} is a conditional attributes set that describes the samples; *D* is a set of classification decision attributes; $V=\cup_{a\in C\cup D}V_{a}$ and *V_a_* is a value set of attribute *a*; *f*: *U* × {*C* ∪ *D*} → *V* is a map function; and *f*(*a*, *x*) represents the value of *x* on attribute *a*
∈
*C* ∪ *D*.

Given a decision system *DS* = <*U*, *C*, *D*> with *B*
⊆
*C*, for any two samples *x*, *y*
∈
*U*, the equivalence relation [49] is described as:(1)IND(B) = {(x, y)|∀a∈B, f(a, x) = f(a, y)}

Then, for any sample *x*
∈
*U*, [*x*]*_B_* = {*y*|*y*
∈
*U*, (*x*, *y*) ∈
*IND*(*B*)} is an equivalence class of *x*, and *U*/*IND*(*B*) (*U*/*B* for short) is called as a partition that is composed of the equivalence classes.

The equivalence class defines two classical sets, named upper and lower approximation sets, as the elementary units [49]. Given a decision system *DS* = <*U*, *C*, *D*> with *B*
⊆
*C* and *X*
⊆
*U*, the upper approximation set and the lower approximation set of *X* with respect to *B* can be described, respectively, as:(2)$$\bar{B}(X)=\{x|{\left[x\right]}_{B}\cap X\neq \emptyset ,x\in U\},$$
(3)$$\underline{B}(X)=\{x|{\left[x\right]}_{B}\subseteq X,x\in U\}.$$

The precision is the ratio of the lower approximation set and the upper approximation set to measure the imprecision of a rough set. The roughness is an inverse of precision by a subtraction in the following.

Given a decision system *DS* = <*U*, *C*, *D*> with *B*
⊆
*C* and *X*
⊆
*U*, the approximation precision of *X* with respect to *B* is described as:(4)$$\rho_{B}(X)=\frac{\left|\underline{B}(X)\right|}{\left|\bar{B}(X)\right|}.$$

Given a decision system *DS* = <*U*, *C*, *D*> with *B*
⊆
*C* and *X*
⊆
*U*, the approximation roughness of *X* with respect to *B* is described as:(5)$$\gamma_{B}(X)=1-\rho_{B}(X)=1-\frac{\left|\underline{B}(X)\right|}{\left|\bar{B}(X)\right|}.$$

The approximation precision and the approximation roughness are used to measure the uncertainty and evaluate a rough set of information systems [41].

### 2.2. Information Entropy Measures

Given a decision system *DS* = <*U*, *C*, *D*> with *B*⊆*C* and *U*/*B* = {*X*_1_, *X*_2_, ⋯, *X_n_*}, the information entropy [50] of *B* is described as:(6)$$H(B)=-\sum_{i=1}^{n}p(X_{i})\log p(X_{i}),$$
where $p\left(X_{i}\right)=\frac{\left|X_{i}\right|}{\;\left|U\right|}$ is the probability of *X_i_*
⊆
*U*/*B*, and |*X_i_*| denotes the cardinality of the equivalence class *X_i_*.

Given a decision system *DS* = <*U*, *C*, *D*> with *B*_1_, *B*_2_
⊆
*C*, *U*/*B*_1_ = {*X*_1_, *X*_2_, ⋯, *X_n_*}, and *U*/*B*_2_ = {*Y*_1_, *Y*_2_, ⋯, *Y_m_*}, then the joint entropy [50] of *B*_1_ and *B*_2_ is defined as:(7)$$H(B_{1}\cup B_{2})=-\sum_{i=1}^{n}\sum_{j=1}^{m}p(X_{i}\cap Y_{j})\log p(X_{i}\cap Y_{j}),$$
where $p\left(X_{i}\cap Y_{j}\right)=\frac{\left|X_{i}\cap Y_{j}\right|}{\;\left|U\right|}$, *i* = 1, 2, ⋯, *n*, and *j* = 1, 2, ⋯, *m*.

Given a decision system *DS* = <*U*, *C*, *D*> with *B*_1_, *B*_2_
⊆
*C*, *U*/*B*_1_ = {*X*_1_, *X*_2_, ⋯, *X_n_*}, and *U*/*B*_2_ = {*Y*_1_, *Y*_2_, ⋯, *Y_m_*}, then the conditional information entropy [50] of *B*_2_ with respect to *B*_1_ is defined as:(8)$$H(B_{2}|B_{1})=-\sum_{i=1}^{n}p(X_{i})\sum_{j=1}^{m}p(Y_{j}|X_{i})\log p(Y_{j}|X_{i}),$$
where $p\left(Y_{j}|X_{i}\right)=\frac{\left|Y_{j}\cap X_{i}\right|}{\;\left|X_{i}\right|}$, *i* = 1, 2, ⋯, *n*, and *j* = 1, 2, ⋯, *m*.

### 2.3. Neighborhood Rough Sets

Given a real-value data set, which is formalized as a neighborhood decision system *NDS* = <*U*, *C*, *D*, *V*, *f*, ∆, *δ*>, where *U* = {*x*_1_, *x*_2_, ⋯, *x**_l_*} is a sample set named universe; *C* = {*a*_1_, *a*_2_, ⋯, *a**_s_*} is a set of all conditional attributes; *D* is a decision attributes set; $V=\cup_{a\in C\cup D}V_{a}$ and *V_a_* is a value set of attribute *a*; *f*: *U* × { *C* ∪ *D* } → *V* is a map function; ∆ → [0, ∞] is a distance function; and *δ* is a neighborhood parameter with 0 ≤ *δ* ≤ 1. In the following, *NDS* = <*U*, *C*, *D*, *V*, *f*, ∆, *δ*> is simply noted as *NDS* = <*U*, *C* ∪ *D*, *δ*>.

Given a neighborhood decision system *NDS* = <*U*, *C* ∪ *D*, *δ*> with *B*
⊆
*C*, a distance function ∆ → [0, ∞], and a neighborhood parameter *δ*
∈ [0, 1], then the neighborhood relation [22] is described as:(9)$$NR_{\delta }(B)=\{(x,y)\in U\times U|\Delta_{B}(x,y)\leq \delta \}.$$

According to the definition of neighborhood relation, for any *x*
∈
*U*, the neighborhood class of *x* with respect to *B*
⊆
*C* is expressed as:(10)$$n_{B}^{\delta }(x)=\{y|x,y\in U,\Delta_{B}(x,y)\leq \delta \}.$$

Because the Euclidean distance function effectively reflects the basic information of the unknown data [22], it is introduced into this paper, and its formula is expressed as:(11)$$\Delta_{B}(x,y)=\sqrt{\sum_{k=1}^{N}{\left|f(a_{k},x)-f(a_{k},y)\right|}^{2}},$$
where *N* is the cardinality of subset *B*.

Given a neighborhood decision system *NDS* = <*U*, *C*
∪
*D*, *δ*> with *B*
⊆
*C* and *X*
⊆
*U*, the neighborhood upper approximation set and the neighborhood lower approximation set of *X* with respect to *B* are denoted, respectively, as:(12)$$\bar{B}{\left(X\right)}_{\delta }=\{x_{i}|n_{B}^{\delta }(x_{i})\cap X\neq \emptyset ,x_{i}\in U\},$$
(13)$$\underline{B}{\left(X\right)}_{\delta }=\{x_{i}|n_{B}^{\delta }(x_{i})\subseteq X,x_{i}\in U\}.$$

## 3. Attribute Reduction Using Lebesgue and Entropy Measures in Neighborhood Decision Systems

### 3.1. Lebesgue Measure-Based Neighborhood Uncertainty Measures

Aiming at the problem that the existing neighborhood rough set model cannot handle the infinite sets, a neighborhood rough set model combined with Lebesgue measure is proposed, which is on the basis of the neighborhood rough set model and the measure theory. Then, the concept of Lebesgue measure is introduced to extend neighborhood rough sets for an infinite set.

For any *M*-dimensional Euclidean space *R^M^*, let *E* be a point set in *R^M^*, and for an open interval *I_i_* of each column covered *E*, $\overset{\infty }{\underset{i=1}{\cup }}I_{i}⊃E$ holds. Then, the sum of its volume is $\mu \;=\;\sum_{i=1}^{\infty }\left|I_{i}\right|$, and all of *μ* form a bounded below set of numbers. The infimum is called the Lebesgue outer measure of *E*, denoted as *m*^*^(*E*), i.e.,:(14)$$m^{*}(E)=\underset{E\subset \overset{\infty }{\underset{i=1}{\cup }}I_{i}}{\inf}\sum_{i=1}^{\infty }\left|I_{i}\right|.$$

The Lebesgue inner measure can be described as $m_{*}$(*E*) = |*I*| − *m*^*^(*I* – *E*). If $m_{*}$(*E*) = *m*^*^(*E*), then *E* is said to be measurable, denoted as the Lebesgue measure *m*(*E*). When the Lebesgue measure of *U* is 0, it can be shown as the cardinality of *U*, i.e., |*U*|. Here, *m*(*X*) is used uniformly to describe the Lebesgue measure of a set *X* in this paper.

**Definition** **1.**
*Given a neighborhood decision system NDS = <U, C*
∪
*D, δ> with non-empty infinite set U and any B*
⊆
*C, ∆_B_(x, y) is a distance function between two objects, and a neighborhood parameter 0 ≤ δ ≤ 1, for any x, y*
∈
*U, then a Lebesgue measure of neighborhood class with respect to B is defined as*
(15)$$m(n_{B}^{\delta }(x))=m(\{y|x,y\in U,\Delta_{B}(x,y)\leq \delta \}).$$


**Proposition** **1.**
*Given a neighborhood decision system NDS = <U, C*
∪
*D, δ> with non-empty infinite set U, and P, Q*
⊆
*C, for any x*
∈
*U, then the following properties hold:*

*(1) m(U) = |U|.*
*(2) If Q*⊆*P, then*$m(n_{P}^{\delta }(x))\leq m(n_{Q}^{\delta }(x))$.*(3) If 0 ≤ α ≤ δ ≤ 1, then*$m(n_{P}^{\alpha }(x))\leq m(n_{P}^{\delta }(x))$.
*(4) For any q*
∈
*Q, m(*
$n_{Q}^{\delta }$
*(x)) ≤ m(*
$\cap_{q\in Q}n_{q}^{\delta }$
*(x)).*

*(5) m(*
$n_{P}^{\delta }(x)$
*) ≠ 0 and m(*
$\cup_{x\in U}n_{P}^{\delta }(x)$
*) = m(U).*


**Proof.** (1) This proof is straightforward.(2) Suppose that any attribute subset *Q*, *R*
⊆
*P*
⊆
*C*, there must exist *P* = *Q*
∪
*R*. From Equation (10), one has that $n_{Q}^{\delta }(x)$ = {*y*|*x, y*
∈
*U*, ∆*_Q_*(*x*, *y*) ≤ *δ*} and $n_{Q\cup R}^{\delta }(x)$ ={*y*|*x, y*
∈
*U*, ∆*_Q_*
∪
*_R_*(*x*, *y*) ≤ *δ*}. Then, it can be obtained from Proposition 1 in [20] that ∆*_Q_*(*x*, *y*) ≤ $\Delta_{Q_{\cup }}$
*_R_*(*x*, *y*), i.e., ∆*_Q_*(*x*, *y*) ≤ ∆*_P_*(*x*, *y*). It follows from Equation (10) that $n_{P}^{\delta }(x)\subseteq n_{Q}^{\delta }(x)$. Therefore, by Equation (15), $m(n_{P}^{\delta }(x))\leq m(n_{Q}^{\delta }(x))$ holds. (3) For any 0 ≤ *α* ≤ *δ* ≤ 1, it follows immediately from Proposition 1 in [41] that $n_{P}^{\alpha }(x)\subseteq n_{P}^{\delta }(x)$ holds. Hence, one has $m(n_{P}^{\alpha }(x))\leq m(n_{P}^{\delta }(x))$.(4) Suppose that any *q*
∈
*Q*
⊆
*C*, it follows from Proposition 1 in [41] that $n_{Q}^{\delta }(x)\subseteq \cap_{q\in Q}n_{q}^{\delta }(x)$. Then, one has that $m(n_{Q}^{\delta }(x))\leq m(\cap_{q\in Q}n_{q}^{\delta }(x))$.(5) For any *P*
⊆
*C*, it can be obtained from Proposition 1 in [41] that $n_{P}^{\delta }(x)\neq \emptyset$ and $\cup_{x\in U}n_{P}^{\delta }(x)$ = *U*. Hence, both $m(n_{P}^{\delta }(x))\neq 0$ and *m*($\cup_{x\in U}n_{P}^{\delta }(x)$) = *m*(*U*) hold. □

**Definition** **2.**
*Given a neighborhood decision system NDS = <U, C*
∪
*D, δ> with non-empty infinite set U, B*
⊆
*C and X*
⊆
*U, an upper approximation and a lower approximation of X with respect to B based on Lebesgue measure are defined, respectively, as:*
(16)$$m(\bar{B}{\left(X\right)}_{\delta })=m(\{x|n_{B}^{\delta }(x)\cap X\neq \emptyset ,x\in U\}),$$
(17)$$m(\underline{B}{\left(X\right)}_{\delta })=m(\{x|n_{B}^{\delta }(x)\subseteq X,x\in U\}).$$


**Proposition** **2.***Given a neighborhood decision system NDS = <U, C*∪*D, δ> with non-empty infinite set U, B*⊆*C and X, Y*⊆ U, then the following properties hold:*(1)*$m(\underline{B}{\left(X\right)}_{\delta })$*≤*$m(\bar{B}{\left(X\right)}_{\delta })$. *(2)*$m(\bar{B}{\left(X\cup Y\right)}_{\delta })=m(\bar{B}{\left(X\right)}_{\delta })+m(\bar{B}{\left(Y\right)}_{\delta })$.*(3)*$m(\underline{B}{\left(X\cup Y\right)}_{\delta })\geq m(\underline{B}{\left(X\right)}_{\delta })+m(\underline{B}{\left(Y\right)}_{\delta })$.*(4) X*⊆*Y*$\Rightarrow m(\underline{B}{\left(X\right)}_{\delta })\leq m(\underline{B}{\left(Y\right)}_{\delta })$.*(5) X*⊆*Y*$\Rightarrow m(\bar{B}{\left(X\right)}_{\delta })\leq m(\bar{B}{\left(Y\right)}_{\delta })$.

**Proof.** (1) This proof is straightforward.(2) Suppose that any $x\in \bar{B}{\left(X\cup Y\right)}_{\delta }$, it follows from Equation (12) that $n_{B}^{\delta }$ (*x*) ∩ (*X* ∪ *Y*) ≠ Ø, and then ($n_{B}^{\delta }$ (*x*) ∩
*X*) ∪ ($n_{B}^{\delta }$ (*x*) ∩
*Y*) ≠ Ø. It is obvious that $n_{B}^{\delta }$ (*x*) ∩
*X* ≠ Ø or $n_{B}^{\delta }$ (*x*) ∩
*Y* ≠ Ø. From Equation (12), one has that *x*
$\in \bar{B}{\left(X\right)}_{\delta }$ or *x*
$\in \bar{B}{\left(Y\right)}_{\delta }$, and then $x\in \bar{B}{\left(X\right)}_{\delta }\cup \bar{B}{\left(Y\right)}_{\delta }$. Thus, it can be obtained that $\bar{B}{\left(X\cup Y\right)}_{\delta }=\bar{B}{\left(X\right)}_{\delta }\cup \bar{B}{\left(Y\right)}_{\delta }$. Therefore, one has *m*($\bar{B}$ (*X*
∪
*Y*)*_δ_*) = *m*($\bar{B}$ (*X*)*_δ_*) + *m* ($\bar{B}$ (*Y*)*_δ_*).(3) Since there exist *X*
⊆
*X*
∪
*Y*
⊆
*U* and *Y*
⊆
*X*
∪
*Y*
⊆
*U*, and it follows from Equation (5) in [49] that $\underline{B}{\left(X\right)}_{\delta }$
⊆
$\underline{B}{\left(X\cup Y\right)}_{\delta }$ and $\underline{B}{\left(Y\right)}_{\delta }$
⊆
$\underline{B}{\left(X\cup Y\right)}_{\delta }$, which yields $\underline{B}$ (*X*)*_δ_*
∪
$\underline{B}$ (*Y*)*_δ_*
⊆
$\underline{B}$ (*X*
∪
*Y*)*_δ_*. Obviously, *m*($\underline{B}$ (*X*)*_δ_*
∪
$\underline{B}$ (*Y*)*_δ_*) ≤ *m* ($\underline{B}$ (*X*
∪
*Y*)*_δ_*) such that one has *m*($\underline{B}$ (*X*)*_δ_*) + *m*($\underline{B}$(*Y*)*_δ_*) ≤ *m*($\underline{B}$(*X*
∪
*Y*)*_δ_*). Hence, *m*($\underline{B}$(*X*
∪
*Y*)*_δ_*) ≥ *m*($\underline{B}$ (*X*)*_δ_*) + *m*($\underline{B}$ (*X*)*_δ_*) can be obtained. (4) Suppose that *X*
⊆
*Y*, it follows that *X*
∩
*Y* = *X*, and then $\underline{B}(X\cap Y)\;=\;\underline{B}(X)$. Similar to the Equation (5) in [49], it can be obtained that $\underline{B}(X)\cap \underline{B}(Y)\;=\;\underline{B}(X)$ in rough sets. Obviously, $\underline{B}{\left(X\cap Y\right)}_{\delta }\;=\;\underline{B}{\left(X\right)}_{\delta }$, and then $\underline{B}{\left(X\right)}_{\delta }\cap \underline{B}{\left(Y\right)}_{\delta }\;=\;\underline{B}{\left(X\right)}_{\delta }$. One has that $\underline{B}{\left(X\right)}_{\delta }\subseteq \underline{B}{\left(Y\right)}_{\delta }$. Hence, $m(\underline{B}{\left(X\right)}_{\delta })\leq m(\underline{B}{\left(Y\right)}_{\delta })$ holds.(5) For any *X*
⊆
*Y*, it follows that *X*
∪
*Y* = *Y*, and then $\bar{B}{\left(X\cup Y\right)}_{\delta }\;=\;\bar{B}{\left(Y\right)}_{\delta }$. From the proof of (2), one has that $\bar{B}{\left(X\right)}_{\delta }\cup \bar{B}{\left(Y\right)}_{\delta }\;=\;\bar{B}{\left(Y\right)}_{\delta }$. Obviously, $\bar{B}{\left(X\right)}_{\delta }$
⊆
$\bar{B}{\left(Y\right)}_{\delta }$. Thus, $m(\bar{B}{\left(X\right)}_{\delta })\leq m(\bar{B}{\left(Y\right)}_{\delta })$ holds. □

**Definition** **3.**
*Given a neighborhood decision system NDS = <U, C*
∪
*D, δ> with non-empty infinite set U, B*
⊆
*C and X*
⊆
*U, a new neighborhood approximate precision of X with respect to B based on Lebesgue measure is defined as:*
(18)$$\rho_{B}^{\delta }(X)=\frac{m(\underline{B}{\left(X\right)}_{\delta })}{m(\bar{B}{\left(X\right)}_{\delta })}.$$

*In a neighborhood decision system, the measure of neighborhood approximate precision is the percentage of possible correct decisions when classifying objects. It is monotonically increasing with the growth of a conditional attribute.*


**Property** **1.**
*Given a neighborhood decision system NDS = <U, C*
∪
*D, δ> with non-empty infinite set U, B*
⊆
*C and X*
⊆
*U, then there exists 0 ≤*
$\rho_{B}^{\delta }(X)$
*≤ 1.*


**Proof.** Suppose that B ⊆ C and X ⊆ U, it follows from Proposition 2 that $m(\underline{B}{\left(X\right)}_{\delta })\leq m(\bar{B}{\left(X\right)}_{\delta })$ established. Then, it is obvious that $0\leq \frac{m(\underline{B}{\left(X\right)}_{\delta })}{m(\bar{B}{\left(X\right)}_{\delta })}\leq 1$, and from Equation (18), one has that 0 ≤ $\rho_{B}^{\delta }(X)$≤ 1. □

The neighborhood approximation precision $\rho_{B}^{\delta }(X)$ is used to reflect the degree of knowledge of acquiring set X. When $\rho_{B}^{\delta }(X)$ = 1, the B boundary of X is an empty set. At this time, the set X is precisely defined on B. When $\rho_{B}^{\delta }(X)$ < 1, the set X has a non-empty B boundary domain, and the set X is undefined on B. Of course, some other metrics can also be used to define the imprecision of X.

**Definition** **4.**
*Given a neighborhood decision system NDS = <U, C*
∪
*D, δ> with non-empty infinite set U, B*
⊆
*C and X*
⊆
*U, a new neighborhood roughness of X with respect to B based on Lebesgue measure is defined as:*
(19)$$\gamma_{B}^{\delta }(X)=1-\rho_{B}^{\delta }(X)=1-\frac{m(\underline{B}{\left(X\right)}_{\delta })}{m(\bar{B}{\left(X\right)}_{\delta })}.$$

*The neighborhood roughness of X with respect to B is opposite and complementary to the neighborhood approximate precision. It represents the degree of incompleteness of obtaining knowledge of set X.*


**Property** **2.**
*Given a neighborhood decision system NDS = <U, C*
∪
*D, δ> with non-empty infinite set U, B*
⊆
*C and X*
⊆
*U, then there exists 0 ≤*
$\gamma_{B}^{\delta }(X)$
*≤ 1.*


**Proof.** It follows immediately from Property 1 that the proof is straightforward. □

**Proposition** **3.***Given a neighborhood decision system NDS = <U, C*∪*D, δ> with non-empty infinite set U, Q*⊆*P*⊆*C and X*⊆*U, then one has*$\rho_{Q}^{\delta }(X)$*≤*$\rho_{P}^{\delta }(X)$*and*$\gamma_{Q}^{\delta }(X)$*≥*$\gamma_{P}^{\delta }(X)$.

**Proof.** Suppose that for any Q ⊆ P ⊆ C and x ∈ U, it follows from Proposition 1 that $m(n_{P}^{\delta }(x))\leq$
$m(n_{Q}^{\delta }(x))$. By Equation (12), $\bar{Q}{\left(X\right)}_{\delta }=\{x|n_{Q}^{\delta }(x)\cap X\neq \emptyset ,x\in U\}$ and $\bar{P}{\left(X\right)}_{\delta }=\{x|n_{P}^{\delta }(x)\cap X\neq \emptyset ,x\in U\}$ hold. Similarly, from Equation (13), one has $\underline{Q}{\left(X\right)}_{\delta }=\{x|n_{Q}^{\delta }(x)\subseteq X,x\in U\}$ and $\underline{P}{\left(X\right)}_{\delta }=\{x|n_{P}^{\delta }(x)\subseteq X,x\in U\}$. It can be obtained that $\underline{Q}{\left(X\right)}_{\delta }\subseteq \underline{P}{\left(X\right)}_{\delta }$ and $\bar{Q}{\left(X\right)}_{\delta }\subseteq \bar{P}{\left(X\right)}_{\delta }$. It follows from Proposition 2 that $m(\underline{Q}{\left(X\right)}_{\delta })\leq m(\underline{P}{\left(X\right)}_{\delta })$ and $m(\bar{Q}{\left(X\right)}_{\delta })\leq m(\bar{P}{\left(X\right)}_{\delta })$. Then, it is obvious that $\frac{m(\underline{Q}{\left(X\right)}_{\delta })}{m(\bar{Q}{\left(X\right)}_{\delta })}\leq \frac{m(\underline{P}{\left(X\right)}_{\delta })}{m(\bar{P}{\left(X\right)}_{\delta })}$ so that one has $1-\frac{m(\underline{Q}{\left(X\right)}_{\delta })}{m(\bar{Q}{\left(X\right)}_{\delta })}\geq 1-\frac{m(\underline{P}{\left(X\right)}_{\delta })}{m(\bar{P}{\left(X\right)}_{\delta })}$. Therefore, both $\rho_{Q}^{\delta }(X)$ ≤ $\rho_{P}^{\delta }(X)$ and $\gamma_{Q}^{\delta }(X)$ ≥ $\gamma_{P}^{\delta }(X)$ hold. □

It is known that the classical measurement methods are used to estimate a set of data classified in a knowledge system [3]. From Proposition 3, the neighborhood approximate precision and the neighborhood roughness are used to measure the uncertainty of rough classification in neighborhood decision systems. 

### 3.2. Neighborhood Entropy-Based Uncertainty Measures

In rough set theory, information entropy is used as a measure to assess the value of equivalence class in a discrete decision system [50]. However, it is not appropriate to measure the neighborhood classes in real-value data sets [18]. To solve this problem, the concept of neighborhood has been introduced into information entropy to extend Shannon entropy [51]. But, most of neighborhood entropy-based measures and their variations only analyze finite sets, which would limit the practical application of neighborhood rough sets to a certain degree. It is known that the Lebesgue measure can measure the uncertainty of the infinite sets [25]. Then, the Lebesgue measure is introduced to study the uncertainty measures of infinite sets in neighborhood decision systems.

Given a neighborhood decision system *NDS* = <*U*, *C*
∪
*D*, *δ*> with *B*
⊆
*C*, $n_{B}^{\delta }(x_{i})$ is a neighborhood class of *x_i_*
∈
*U*, and then Hu et al. [21] described the neighborhood entropy of *x_i_* with respect to *B* as follows:(20)$$H_{\delta }^{x_{i}}(B)=-\log\frac{\left|n_{B}^{\delta }(x_{i})\right|}{\left|U\right|}.$$

**Definition** **5.**
*Given a neighborhood decision system NDS = <U, C*
∪
*D, δ> with non-empty infinite set U, B*
⊆
*C and x_i_*
∈
*U, a new neighborhood entropy of B based on Lebesgue measure is defined as:*
(21)$$H_{\delta }^{x_{i}}(B)=-\log_{2}\frac{m(n_{B}^{\delta }(x_{i}))}{m(U)}.$$


**Definition** **6.***Given a neighborhood decision system NDS* = <U, C ∪
*D, δ> with non-empty infinite set U, and B*
⊆
*C, an average neighborhood entropy of the universe U based on Lebesgue measure is defined as:*(22)$$H_{\delta }(B)=-\frac{1}{m(U)}\sum_{i=1}^{\left|U\right|}\log_{2}\frac{m(n_{B}^{\delta }(x_{i}))}{m(U)}.$$

**Proposition** **4.**
*Given a neighborhood decision system NDS = <U, C*
∪
*D, δ > with non-empty infinite set U and P, Q*
⊆
*C, for any x_i_*
∈
*U, if*
$n_{P}^{\delta }(x_{i})=n_{Q}^{\delta }(x_{i})$
*, then H_δ_(P) = H_δ_(Q).*


**Proof.** Suppose that for any *x_i_*
∈
*U*, $n_{P}^{\delta }(x_{i})=n_{Q}^{\delta }(x_{i})$, and it follows that from Equation (15) that $m(n_{P}^{\delta }(x_{i}))=$
$m(n_{Q}^{\delta }(x_{i}))$. Then, it can be obtained that $\sum_{i=1}^{\left|U\right|}\log_{2}\frac{m(n_{P}^{\delta }(x_{i}))}{m(U)}=\sum_{i=1}^{\left|U\right|}\log_{2}\frac{m(n_{Q}^{\delta }(x_{i}))}{m(U)}$. Hence, by Equation (22), one has *H_δ_*(*P*) = *H_δ_*(*Q*). □

**Definition** **7.**
*Given a neighborhood decision system NDS = <U, C*
∪
*D, δ> with non-empty infinite set U and B*
⊆
*C,*
$n_{B}^{\delta }(x_{i})$
*is the neighborhood class of x_i_ in neighborhood relation, and [x_i_]_d_ is an equivalence class formed by the decision attribute d of x_i_ in equivalence relation. Then, a joint entropy of subsets B and d based on Lebesgue measure is defined as:*
(23)$$H_{\delta }(Bd)=-\frac{1}{m(U)}\sum_{i=1}^{\left|U\right|}\log_{2}(\frac{m(n_{B}^{\delta }(x_{i})\cap {\left[x_{i}\right]}_{d})}{m(U)}).$$


**Proposition** **5.**
*Given a neighborhood decision system NDS = <U, C*
∪
*{d}, δ> with non-empty infinite set U, and Q*
⊆
*P*
⊆
*C, then H_δ_(Qd) ≤ H_δ_(Pd).*


**Proof** Suppose that any *Q*
⊆
*P*
⊆
*C*, it can be obtained from Proposition 1 that $n_{P}^{\delta }(x)\subseteq n_{Q}^{\delta }(x)$ and $m(n_{P}^{\delta }(x))\leq m(n_{Q}^{\delta }(x))$. Let [*x_i_*]*_d_* be an equivalence class formed by *d* of *x_i_* in equivalence relation. Then, $\{x_{i}\}\subseteq n_{P}^{\delta }(x_{i})\cap {\left[x_{i}\right]}_{d}\subseteq n_{Q}^{\delta }(x_{i})\cap {\left[x_{i}\right]}_{d}\subseteq U$ holds. Obviously, one has that $m(\{x_{i}\})\leq m(n_{P}^{\delta }(x_{i})\cap {\left[x_{i}\right]}_{d})\leq$
$m(n_{Q}^{\delta }(x_{i})\cap {\left[x_{i}\right]}_{d})\leq m(U)$. It follows that $\frac{1}{m(U)}\leq \frac{m(n_{P}^{\delta }(x_{i})\cap {\left[x_{i}\right]}_{d})}{m(U)}\leq \frac{m(n_{Q}^{\delta }(x_{i})\cap {\left[x_{i}\right]}_{d})}{m(U)}\leq 1$. Then, it is obvious that $\log_{2}(\frac{1}{m(U)})\leq \log_{2}(\frac{m(n_{P}^{\delta }(x_{i})\cap {\left[x_{i}\right]}_{d})}{m(U)})\leq \log_{2}(\frac{m(n_{Q}^{\delta }(x_{i})\cap {\left[x_{i}\right]}_{d})}{m(U)})\leq 0$. Thus, it can be obtained that $0\leq -\frac{1}{m(U)}\sum_{i=1}^{\left|U\right|}\log_{2}(\frac{m(n_{Q}^{\delta }(x_{i})\cap {\left[x_{i}\right]}_{d})}{m(U)})\leq -\frac{1}{m(U)}\sum_{i=1}^{\left|U\right|}\log_{2}(\frac{m(n_{P}^{\delta }(x_{i})\cap {\left[x_{i}\right]}_{d})}{m(U)})\leq \log_{2}m(U)$. Therefore, *H_δ_*(*Qd*) ≤
*H_δ_*(*Pd*) holds.  □

**Definition** **8.**
*Given a neighborhood decision system NDS = <U, C*
∪
*{d}, δ> with non-empty infinite set U and B*
⊆
*C, for any class of object x*
∈
*U with respect to B,*
$n_{B}^{\delta }(x)$
*is a neighborhood class of x generated by the neighborhood relation NR_δ_(B), [x_i_]_d_ is an equivalence class of x_i_ generated by equivalence relation IND(d), and U/{d} = {d_1_, d_2_, …, d_t_,…}. Then, the neighborhood roughness joint entropy based on Lebesgue measure of d with respect to B is defined as:*
(24)$$NRH(d,B)=-\frac{1}{m(U)}\sum_{j=1}^{\infty }\log_{2}{}^{\left(2-\gamma_{B}^{\delta }(d_{j})\right)}\times \int_{x_{i}\in U}\log_{2}(\frac{m(n_{B}^{\delta }(x_{i})\cap d_{j})}{m(U)})dx.$$


It is noted that Wang et al. [46] stated that all conceptions and computations in rough set theory based on the upper and lower approximation sets are called the algebra view of the rough set theory, and the notions of information entropy and its extensions are called the information view of rough sets. It follows from Equation (24) that $\gamma_{B}^{\delta }(d_{i})$ is the neighborhood roughness of *d_i_* with respect to *B* in the algebra view and it represents the degree of incompleteness of obtaining knowledge of set *d_i_*, and $\sum_{j=1}^{\infty }\int_{x_{i}\in U}\log_{2}(\frac{m(n_{B}^{\delta }(x_{i})\cap d_{j})}{m(U)})dx$ is the definition of joint entropy in the information view. Hence, Definition 8 can efficiently analyze and measure the uncertainty of neighborhood decision systems based on Lebesgue and entropy measures from both the algebra view and the information view. 

**Property** **3.***Given a neighborhood decision system NDS* = <U, C ∪
*{d}, δ> with non-empty infinite set U, B*
⊆
*C, and U/{d} = {d_1_, d_2_, …, d_t_,…}, then*
$$NRH(d,B)=-\frac{1}{\left|U\right|}\sum_{j=1}^{\infty }\log_{2}{}^{\left(2-\gamma_{B}^{\delta }(d_{j})\right)}\times \int_{x_{i}\in U}\log_{2}(\frac{m(n_{B}^{\delta }(x_{i})\cap d_{j})}{\left|U\right|})dx\geq 0$$

**Proof.** Suppose that for any *B*
⊆
*C*, it follows from Proposition 1 that *m*(*U*) = |*U*|. Then, one has that $NRH(d,B)=-\frac{1}{\left|U\right|}\sum_{j=1}^{\infty }\log_{2}{}^{\left(2-\gamma_{B}^{\delta }(d_{j})\right)}\times \int_{x_{i}\in U}\log_{2}(\frac{m(n_{B}^{\delta }(x_{i})\cap d_{j})}{\left|U\right|})dx$. As known from Equation (23), due to *m*($n_{B}^{\delta }(x_{i})\cap d_{j})$ ≤ m(*U*), it is obvious that $\log_{2}(\frac{m(n_{B}^{\delta }(x_{i})\cap d_{j})}{\left|U\right|})$ ≤ 0 and $\log_{2}{}^{\left(2-\gamma_{B}^{\delta }(d_{i})\right)}$ ≥ 0. Therefore, *NRH*(*d*, *B*) ≥ 0 holds. □

**Proposition** **6.***Given a neighborhood decision system NDS* = <U, C ∪
*D, δ> with non-empty infinite set U, P*
⊆
*Q*
⊆
*C, and U/{d} = {d_1_, d_2_, …, d_t_,…}, then NRH(D, Q) ≤ NRH(D, P).*

**Proof.** Suppose that for any *Q*
⊆
*P*
⊆
*C* and *X*
⊆
*U*, it follows from Proposition 3 that $\rho_{Q}^{\delta }(X)\leq \rho_{P}^{\delta }(X)$ and $\gamma_{P}^{\delta }(X)$ ≤ $\gamma_{Q}^{\delta }(X)$. It is clear that $0\leq \log^{\left(2-\gamma_{Q}^{\delta }(d_{i})\right)}\leq \log^{\left(2-\gamma_{P}^{\delta }(d_{i})\right)}$. From Proposition 5, one has that *H_δ_*(*Qd*) ≤ *H_δ_*(*Pd*). Then, it can be obviously obtained that $-\frac{1}{m(U)}\int_{x_{i}\in U}\log_{2}(\frac{m(n_{Q}^{\delta }(x_{i})\cap d_{j})}{\left|U\right|})dx\leq -\frac{1}{m(U)}$
$\int_{x_{i}\in U}\log_{2}(\frac{m(n_{P}^{\delta }(x_{i})\cap d_{j})}{\left|U\right|})dx$. When $n_{Q}^{\delta }(x)=n_{P}^{\delta }(x)$ for any *x*
∈
*U*, one has $\gamma_{P}^{\delta }(d_{j})=\gamma_{Q}^{\delta }(d_{j})$ and $n_{Q}^{\delta }(x)\cap d_{j}$
$=n_{P}^{\delta }(x)\cap d_{j}$, where 1 ≤ j. Thus, *NRH*(*D*, *Q*) = *NRH*(*D*, *P*). Therefore, *NRH*(*D*, *Q*) ≤ *NRH*(*D*, *P*) holds. □

The monotonicity is one of the most important properties for an effective uncertainty measure of attribute reduction. According to Proposition 6, it is quite obvious that the neighborhood roughness joint entropy is monotonically increasing when adding the conditional attributes, which validates the monotonicity of the proposed uncertainty measure. Furthermore, the monotonicity contributes to the selection of the greedy method for attribute reduction.

**Definition** **9.**
*Given a neighborhood decision system NDS = <U, C*
∪
*D, δ> with non-empty infinite set U and B*
⊆
*C, for any a*
∈
*B, then the internal attribute significance of a in B relative to D is defined as:*
*Sig^inner^*(*a*, *B*, *D*) = *NRH*(*D*, *B*) − *NRH*(*D*, *B* − {*a*}).(25)


**Definition** **10.**
*Given a neighborhood decision system NDS = <U, C*
∪
*D, δ> with non-empty infinite set U and B*
⊆
*C, for any a*
∈
*C − B, then the external attribute significance of a relative to D is defined as:*
(26)$${Sig}^{outer}(a,\;B,\;D)\;=\;NRH(D,\;B\cup \{a\})\;-\;NRH(D,\;B)$$


**Definition** **11.**
*Given a neighborhood decision system NDS = <U, C*
∪
*D, δ> with non-empty infinite set U and any a*
∈
*C, if NRH(D, C) > NRH(D, C − {a}), that is, Sig^inner^(a, C, D) > 0, then the attribute a is a core of C relative to D.*


**Definition** **12.**
*Given a neighborhood decision system NDS = <U, C*
∪
*D, δ> with non-empty infinite set U and B*
⊆
*C, if any a*
∈
*B is necessary in B if and only if Sig^inner^(a, B, D) > 0; otherwise a is unnecessary. If each a*
∈
*B is necessary, one can say that B is independent; otherwise B is dependent.*


**Definition** **13.**
*Given a neighborhood decision system NDS = <U, C*
∪
*D, δ> with non-empty infinite set U and B*
⊆
*C, if NRH(D, B) = NRH(D, C), and for any a*
∈
*B, there exists NRH(D, B) > NRH(D, B − {a}), then it is said that B is a reduct of C relative to D.*


### 3.3. Comparative Analysis with Two Representative Reducts

It is known that the definition of reducts from the algebra view is usually equivalent to its definition from the information view in a general information system. What’s more, the relative reduct of a decision system in the information view includes that in the algebra view. Thus, Wang et al. [46] declared that any relative reduct of a decision system in the information view must be its relative reduct in the algebra view, so that some heuristic algorithms can be designed further using this conclusion. Based on the ideas of the classification in [46], the definition of reducts based on neighborhood roughness joint entropy of a neighborhood decision system should be developed from the algebra view and the information view in neighborhood rough set theory. For convenience, the reduct in Definition 13 is named as the neighborhood entropy reduct. Liu et al. [38] presented a reduct based on positive region in the neighborhood decision system similar with classical rough set model. This representative relative reduct based on positive region is called the algebra view of the neighborhood rough set theory. Chen et al. [20] defined information quantity similar to information entropy to evaluate the neighborhood classes, used the joint entropy gain to evaluate the significance of a selecting attribute, and proposed a representative joint entropy gain-based reduction algorithm, which is called a reduct in the information view of neighborhood rough sets.

Given a neighborhood decision system *NDS* = <*U*, *C*
∪
*D*, *δ*> with non-empty infinite set *U*, *B*
⊆
*C* and *D* = {*d*}. Then, a positive region reduct of the neighborhood decision system is presented as follows in [38]: for any *a*
∈
*B*, if |*POS_B_*(*D*)| = |*POS_C_*(*D*)| and |*POS_B_*
_− {*a*}_(*D*)| < |*POS_B_*(*D*)|, where *POS_B_*(*D*) = $\cup \{\underline{B}{\left(X\right)}_{\delta }|X\in U/D\}$ is the positive region of *D* with respect to *B*, *B* is a relative reduct of the neighborhood decision system.

**Proposition** **7.**
*Given a neighborhood decision system NDS = <U, C*
∪
*D, δ> with non-empty infinite set U, and B*
⊆
*C, if B is a neighborhood entropy reduct of the neighborhood decision system, then B is a positive region reduct of the neighborhood decision system.*


**Proof.** Let *U* = {*x*_1_, *x*_2_, …, *x_n_*, …}, and *U*/*D* = {*d*_1_, *d*_2_, …, *d_t_, …*}. Suppose that for a subset *B*⊆
*C*, it follows from Definition 13 that if *NRH*(*D*, *B*) = *NRH*(*D*, *C*), and for any *a*
∈
*B*, there exists *NRH*(*D*, *B*) > *NRH*(*D*, *B* − {*a*}), then *B* is a neighborhood entropy reduct of *C* relative to *D*. When *NRH*(*D*, *B*) = *NRH*(*D*, *C*), it can be obtained from Proposition 6 that $n_{B}^{\delta }(x)=n_{C}^{\delta }(x)$, $\gamma_{B}^{\delta }(d_{j})=\gamma_{C}^{\delta }(d_{j})$ and $n_{B}^{\delta }(x)\cap d_{j}$
$=n_{C}^{\delta }(x)\cap d_{j}$ hold, where any *x*
∈
*U* and 1 ≤ j. By Equation (13), one has that $\underline{B}{\left(D\right)}_{\delta }=\underline{C}{\left(D\right)}_{\delta }$. So it is obvious that *POS_B_*(*D*) = *POS_C_*(*D*), i.e., |*POS_B_*(*D*)| = |*POS_C_*(*D*)|. For any *a*
∈
*B*, *B* − {*a*} ⊂
*B*, and from Theorem 1 in [38], one has that $\underline{B-\{a\}}{\left(D\right)}_{\delta }\subseteq \underline{B}{\left(D\right)}_{\delta }$, so that *POS_B_*
_− {*a*}_(*D*)⊆
*POS_B_*(*D*) holds. Because for any *a*
∈
*B*, there exists *NRH*(*D*, *B*) > *NRH*(*D*, *B* − {*a*}), thus $\underline{B-\{a\}}{\left(D\right)}_{\delta }\subset \underline{B}{\left(D\right)}_{\delta }$ holds. It follows that *POS_B_*
_− {*a*}_(*D*) ⊂
*POS_B_*(*D*). Thus, |*POS_B_*
_− {*a*}_(*D*)| < |*POS_B_*(*D*)| for any *a*
∈
*B*. Therefore, *B* is a positive region reduct of the neighborhood decision system. □

Notably, the inverse relation of this proposition generally does not hold. According to the above discussions, Proposition 7 shows that the definition of the neighborhood entropy reduct includes that of positive region reduct in the algebra view.

Given a neighborhood decision system *NDS* = <*U*, *C*
∪
*D*, *δ*> with non-empty infinite set *U*, *B*
⊆
*C* and *D* = {*d*}. For any *a*
∈
*B*, a reduct of the neighborhood decision system, named as the entropy gain reduct is proposed in [20] as follows: if *H*(*Bd*) = *H*(*Cd*) and *H*({*B* − {*a*}}*d*) < *H*(*Bd*), where $H(Bd)=-\frac{1}{\left|U\right|}\sum_{i=1}^{\left|U\right|}\log_{2}(\frac{n_{B}^{\delta }(x_{i})\cap {\left[x_{i}\right]}_{d}}{\left|U\right|})$ describes the joint entropy of *B* and *d*, *B* is an entropy gain reduct of the neighborhood decision system. 

**Proposition** **8.**
*Given a neighborhood decision system NDS = <U, C*
∪
*D, δ> with non-empty infinite set U and B*
⊆
*C, then B is a neighborhood entropy reduct of the neighborhood decision system if and only if B is an entropy gain reduct of the neighborhood decision system.*


**Proof.** Let *U* = {*x*_1_, *x*_2_, …, *x_n_*, …}, and *U*/*D* = {*d*_1_, *d*_2_, …, *d_t_, …*}. Suppose that for a subset *B*⊆
*C*, it follows from Definition 13 that if *NRH*(*D*, *B*) = *NRH*(*D*, *C*), and for any *a*
∈
*B*, there exists *NRH*(*D*, *B*) > *NRH*(*D*, *B* − {*a*}), then *B* is a neighborhood entropy reduct of *C* relative to *D*. Similar to the proof of Proposition 7, when *NRH*(*D*, *B*) = *NRH*(*D*, *C*), from Proposition 6, one has that $n_{B}^{\delta }(x)\cap d_{j}=$
$n_{C}^{\delta }(x)\cap d_{j}$, where any *x*
∈
*U* and 1 ≤ j. It is obvious that *H*(*Bd*) = *H*(*Cd*). Since *B* − {*a*} ⊂
*B*, from Proposition 2 in [20], one has that *H*({*B* − {*a*}}*d*) ≤ *H*(*Bd*). Because for any *a*
∈
*B*, there exists *NRH*(*D*, *B*) > *NRH*(*D*, *B* − {*a*}), so *H*({*B* − {*a*}}*d*) < *H*(*Bd*) holds. Hence, *B* is an entropy gain reduct of the neighborhood decision system.Suppose that for a subset *B*
⊆
*C*, and any *a*
∈
*B*, if *H*(*Bd*) = *H*(*Cd*) and *H*({*B* − {*a*}}*d*) < *H*(*Bd*), then B is an entropy gain reduct of *C* relative to *D*. Similar to the proof of Proposition 6, when $n_{B}^{\delta }(x)=n_{C}^{\delta }(x)$, by Equations (16), (17) and (19), one has that $\gamma_{B}^{\delta }(d_{j})=\gamma_{C}^{\delta }(d_{j})$, and then it is obvious that $n_{B}^{\delta }(x)\cap d_{j}=n_{C}^{\delta }(x)\cap d_{j}$, where any *x*
∈
*U* and 1 ≤ j. Thus, it can be obtained from Equation (24) that *NRH*(*D*, *B*) = *NRH*(*D*, *C*). Because *B* − {*a*} ⊂
*B*, it follows from Proposition 6 that *NRH*(*D*, *B* − {*a*}) ≤ *NRH*(*D*, *B*). Since for any *a*
∈
*B*, there exists *H*({*B* − {*a*}}*d*) < *H*(*Bd*). So, one has that *NRH*(*D*, *B* − {*a*}) < *NRH*(*D*, *B*). Therefore, *B* is a neighborhood entropy reduct of the neighborhood decision system. □

Proposition 8 shows that in a neighborhood decision system, the neighborhood entropy reduct is equivalent to the entropy gain reduct in the information view. According to Propositions 7 and 8, it can be concluded that the definition of neighborhood entropy reduct includes two representative reducts proposed in the algebra view and the information view. Therefore, the definition of neighborhood entropy reduct denotes a mathematical quantitative measure to evaluate the knowledge uncertainty of different attribute sets in neighborhood decision systems.

### 3.4. Description of the Attribute Reduction Algorithm

In order to facilitate the understanding of the attribute reduction method, the process of attribute reduction algorithm for data classification is illustrated in Figure 1.

To support efficient attribute reduction, an attribute reduction algorithm based on neighborhood roughness joint entropy (ARNRJE) is constructed and described as Algorithm 1.

**Algorithm 1.** ARNRJE**Input:** A neighborhood decision system *NDS* = <*U*, *C*
∪
*D*, *δ*>, and neighborhood parameter *δ*.**Output:** An optimal reduction set *B*.Initialize *B* = *Ø*.Calculate *NRH* (*D*, *C*).For *i* = 1 to |*C*| do // Obtain the attribute core Calculate *Sig^inner^*(*a_i_*, *C*, *D*). If *Sig^inner^*(*a_i_*, *C*, *D*) > 0, then *B*
*=*
*B*
∪ {*a_i_*}.End forLet R = C – B.DoFor *j* = 1 to |*R*|
a)Calculate *NRH*(*D*, *B*
∪ {*a_j_*}).b)Select *a_j_* to make it satisfy max{*a_j_*
∈
*R*|*NRH*(*D*, *B*
∪ {*a_j_*})}, and if there are multiple attributes that satisfy the maximum, then the front should be selected.End forLet *B = B*
∪ {*a_j_*} and *R*
*= R* − {*a_j_*}, and calculate *NRH*(*D*, *B*).While *NRH*(*D*, *B*) ≠ *NRH*(*D*, *C*)For *k* = 1 to |*B*| do // Verify the completeness of the reduction subset Select *a**_k_*
∈
*B*. Compute *NRH*(*D*, *B* − {*a_k_*}). If *NRH*(*D*, *B* − {*a_k_*}) ≥ *NRH*(*D*, *B*), then let *B* = *B* − {*a_k_*}.End forReturn an optimal reduction set *B*.

### 3.5. Complexity Analysis of ARNRJE

In the ARNRJE algorithm, suppose that there are *m* attributes and *n* samples, and then the calculation of neighborhood classes is frequent in neighborhood decision systems. The process of deriving the neighborhood classes has a great influence on the time complexity of selecting attributes. Notably, the main computation of ARNRJE includes two aspects: obtaining neighborhood classes and computing the neighborhood roughness joint entropy. Then, the buckets sorting algorithm [38] is introduced to further reduce the time complexity of neighborhood classes, and the time complexity of computing neighborhood classes should be *O*(*mn*). Meanwhile, the time complexity of calculating neighborhood roughness joint entropy is *O*(*n*). Since *O*(*n*) < *O*(*mn*), the complexity of computing neighborhood roughness joint entropy should be *O*(*mn*). Because there exist two loops at steps 3 and 8 of the ARNRJE algorithm, in the worst case, the time complexity of ARNRJE algorithm is *O*(*m*^3^*n*). As known that, in the process of achieving attribute reduction task, we usually select a litter of attributes. Suppose that the number of selected attributes is *m_R_*, and in the computation of neighborhood classes, we only need to consider the candidate attributes without touching on the whole attribute set. So, the complexity of computing neighborhood classes is decreased to *O*(*m_R_n*). For the ARNRJE algorithm, the times of the outer loop are *m* and the times of the inner loop are *m* – *m_R_*. Thus, the time complexity of the ARNRJE algorithm is *O*(*m_R_n*(*m* – *m_R_*)*m*). It is obvious that *m_R_*
≪
*m* in most cases. Therefore, the time complexity of ARNRJE algorithm is approximately *O*(*mn*). So far, ARNRJE appears to be more efficient than some of the existing algorithms for attribute reduction in [18,36,45,52,53] for neighborhood decision systems. Furthermore, its space complexity is *O*(*mn*).

### 3.6. An Illustrative Example

In the following, the performance of ARNRJE algorithm is shown through an illustrative example in [20]. A neighborhood decision system *NDS* = <*U*, *C*
∪
*D*, *δ* > is employed, where *U* = {*x*_1_, *x*_2_, *x*_3_, *x*_4_}, *C* = {*a*, *b*, *c*}, and *D* = {*d*} with the values {*Y*, *N*}. The neighborhood decision system is shown in Table 1.

For Table 1, we use Algorithm 1 for attribute reduction. Given *δ* = 0.3, the attribute reduction steps are as follows:

(1) Initialize *B* = *Ø*.

(2) Let *B = C* = {*a*, *b*, *c*}, and the Euclidean distance function is used to calculate the distance between any two objects as follows: 

∆*_B_*(*x*_1_, *x*_2_) = 0.54, ∆*_B_*(*x*_1_, *x*_3_) = 0.35, ∆*_B_*(*x*_1_, *x*_4_) = 0.68, ∆*_B_*(*x*_2_, *x*_3_) = 0.16, ∆*_B_*(*x*_2_, *x*_4_) = 0.41, and ∆*_B_*(*x*_3_, *x*_4_) = 0.302.

Then, we can get the following neighborhood classes: 

$n_{B}^{\delta }$(*x*_1_) = {*x*_1_}, $n_{B}^{\delta }$(*x*_2_) = {*x*_2_, *x*_3_}, $n_{B}^{\delta }$ (*x*_3_) = {*x*_2_, *x*_3_}, and $n_{B}^{\delta }$ (*x*_4_) ={*x*_4_}.

Under the equivalence relation, *U*/*d* = {*d*_1_, *d*_2_} = {{*x*_1_, *x*_2_},{*x*_3_, *x*_4_}}, the Euclidean distance function is ∆*_B_*, and then the upper and lower approximation sets of attribute subset *B* about *d*_1_ and *d*_2_ are calculated, respectively, by

$\bar{B}{\left(d_{1}\right)}_{\delta }$ = {*x*_1_, *x*_2_, *x*_3_}, $\underline{B}{\left(d_{1}\right)}_{\delta }$ = {*x*_1_}; $\bar{B}{\left(d_{2}\right)}_{\delta }$ = {*x*_2_, *x*_3_, *x*_4_}, and $\underline{B}{\left(d_{2}\right)}_{\delta }$ = {*x*_4_}. 

Thus, one has that $\rho_{B}^{\delta }(d_{1})=\frac{1}{3}$, $\gamma_{B}^{\delta }(d_{1})=1-\rho_{B}^{\delta }(d_{1})=\frac{2}{3}$, $\rho_{B}^{\delta }(d_{2})=\frac{1}{3}$, and $\gamma_{B}^{\delta }(d_{2})=1-\rho_{B}^{\delta }(d_{2})=\frac{2}{3}$. 

It follows that $NRH(D,C)=-\frac{1}{\left|U\right|}\sum_{j=1}^{2}\log_{2}{}^{\left(2-\gamma_{C}^{\delta }(d_{i})\right)}\times \int_{x_{i}\in U}\log_{2}(\frac{m(n_{C}^{\delta }(x_{i})\cap d_{j})}{\left|U\right|})dx=0.83.$

(3) For all the attributes in *C*, the attribute significance is calculated by 

*sig^inner^*(*a*, *C*, *D*) = 0.5081, *sig^inner^*(*b*, *C*, *D*) = 0, and *sig^inner^*(*c*, *C*, *D*) = 0.2075. 

Since *sig^inner^*(*a*, *C*, *D*) > 0 and *sig^inner^*(*c*, *C*, *D*) > 0, then one has *B* = {*a*, *c*}.

(4) Since *B* = {*a*, *c*} ≠ *Ø*, by computing, then *NRH*(*D*, *B*) = 0.83.

(5) Because *NRH*(*D*, *B*) = *NRH*(*D*, *C*), it is obtained that the reduction set *B* = {*a*, *c*}.

(6) One computes *NRH*(*D*, *B* – {*a*}) = *NRH*(*D*, {*c*}) = 0.3219 and *NRH*(*D*, *B* – {*c*}) = *NRH*(*D*, {*a*}) = 0.2925. Since *NRH*(*D*, *B* – {*a*}) ≤ *NRH*(*D*, *B*) and *NRH*(*D*, *B* – {*c*}) ≤ *NRH*(*D*, *B*), then *B* = {*a*, *c*} holds.

(7) Return the reduction attribute subset *B* = {*a*, *c*}.

## 4. Experimental Results and Analysis

### 4.1. Experiment Preparation

The objective of an attribute reduction method usually includes two aspects: one is to select a small number of attributes and the other is to maintain high classification accuracy. To verify the classification performances of our proposed attribute reduction method described in Subsection 3.4, the comprehensive results of all contrasted algorithms can be achieved and analyzed on nine public data sets (five UCI data sets and four gene expression data sets). The selected five UCI data sets with low-dimensional attributes include the Wine, Sonar, Segmentation, Wdbc, and Wpbc data sets [54]. The selected four gene expression data sets with high-dimensional attributes include the Prostate, DLBCL, Leukemia and Tumors data sets [55]. All the data sets are summarized in Table 2. 

The experiments were performed on a personal computer running Windows 10 with an Intel(R) Core(TM) i5-6500 CPU operating at 3.20 GH, and 4.0 GB memory. All the simulation experiments were implemented in MATLAB 2016a programming software, and two different classifiers (KNN and LibSVM) were selected to verify the classification accuracy in WEKA software, where the parameter k in KNN was set to 3 and the linear kernel functions were selected in LibSVM. All of the following experimental comparisons for classification on the selected attributes are implemented using a 10-fold cross-validation with all the test data sets, where every data set is first randomly divided into ten portions which are the same size subset of data each other, one data subset is used as the testing data set, the rest nine data subsets are used as the training data set, and each of the ten data subsets only is employed exactly once as the testing data set; secondly, the operation of the cross-validation is repeated ten times; finally, the average of ten test results is as the obtained classification accuracy [45].

### 4.2. Effect of Different Neighborhood Parameter Values

The following part of our experiments concerns the reduction rate and the classification accuracy under the different neighborhood parameter values. The reduction rate and the classification accuracy of an attribute subset for the different neighborhood parameter values are discussed to obtain a suitable neighborhood parameter value and a subset of attributes. Chen et al. [20] adopted a reduction rate for evaluating the attribute redundancy degree of attribute reduction algorithms, which is described as:(27)$$Rate_{\delta }=\frac{\left|C\right|-\left|R\right|}{\left|C\right|},$$
where |*C*| is the number of all of the conditional attributes in the data set, and |*R*| is the number of the reduced attributes obtained under the different neighborhood parameter values.

For the four high-dimensional gene expression data sets (Prostate, DLBCL, Leukemia and Tumors), the Fisher score method [8] is used for preliminary dimension reduction. For each gene expression data set, the Fisher score method is used to calculate the Fisher score value and sequence it for each gene, and *l* genes are selected to construct a candidate gene subset. The classification accuracy under different dimensions is obtained by using WEKA software, so that the appropriate dimension can be selected for the attribute reduction algorithm. Figure 2 shows the changing trend of the classification accuracy versus the number of genes on the four gene expression data sets.

From Figure 2, it can be seen that when the number of genes increases, the classification accuracy is also changed. Since the cardinality of selected genes and the classification accuracy for selected genes are all important, they are two indices for evaluating the classification performance of attribute reduction methods. Then, the appropriate values of *l* need to be selected from Figure 2. Hence, the values of *l* are set to 50-dimension and 100-dimension for the Prostate and DLBCL data set, respectively. For the Leukemia and Tumors data sets, the values of *l* can be set to 300-dimension and 200-dimension, respectively.

The classification accuracy of data sets given in Table 2 was obtained by using the ARNRJE algorithm with different neighborhood parameter values. After obtaining the results of attribute reduction with different parameters, WEKA is used to obtain the classification accuracy under the 3NN and LibSVM classifiers. The results are shown in Figure 3, where the horizontal coordinate denotes the neighborhood parameters with *δ*
∈ [0.05, 1] at intervals of 0.05, and the left and the right vertical coordinate represent the classification accuracy and the reduction rate, respectively.

Figure 3 shows that the classification accuracy of selected attributes by the ARNRJE algorithm is increasing, and the reduction rate is decreasing with the neighborhood parameter values changing from 0.05 to 1 in most cases. It is easily observed that the different neighborhood parameter values have great influence on the classification performance of ARNRJE. Then, this illustrates that the thinner the granule is, the smaller the roughness of the granule is when the values of different neighborhood parameter are smaller. It follows that the reduction rate increases as the roughness of the granule decreases. Thus, the optimal neighborhood parameter values can be selected for each data set. Figure 3a shows the classification accuracy of the Wine data set with different neighborhood parameter values, and the neighborhood parameter can be set to 0.65. Figure 3b demonstrates the classification accuracy of the Sonar data set with different neighborhood parameter values. The reduction rate decreases as the neighborhood parameter values increase, and the classification accuracy of selected attributes reaches the relative maximum value when the neighborhood parameter is 0.4. So, the neighborhood parameter of the Sonar data set can be set to 0.4. From Figure 3c, it can be seen that the classification accuracy reaches the relative best performance when the parameter equals 0.75 on the Segmentation data set. Figure 3d displays the classification accuracy of the Wdbc data set for the different neighborhood parameters. The neighborhood parameter of the Wdbc data set can be set to 0.35. Similar to Figure 3b, Figure 3e reveals the classification accuracy of selected attributes reaching a relative maximum when the parameter is set as 0.55 for Wpbc data set. For Prostate and DLBCL with the different neighborhood parameter values in Figure 3f,g, the neighborhood parameter of the Prostate data set can be set to 0.9, and that of the DLBCL data set should be set to 0.5. Figure 3h,i demonstrates that the reduction rate is decreasing as the parameters increase in most situations, and the neighborhood parameter of Leukemia and Tumors data sets can be set to 0.4 and 0.25, respectively.

### 4.3. Comparison of Reduction Results with Three Related Reduction Algorithms

This portion of our experiments evaluates the performance of our proposed ARNRJE algorithm in terms of the selected attribute subset of data sets. The ARNRJE algorithm is compared with the following two reduction algorithms: (1) the fuzzy information entropy-based feature selection algorithm (FINEN) [52,58], and (2) the neighborhood entropy-based feature selection algorithm (NEIEN) [18]. By using the neighborhood parameters where the classification accuracy is obtained in Subsection 4.2, the attribute reduction results and the number of selected attributes on the nine data sets from Table 2 are shown in Table 3.

Table 3 lists the selected attribute subsets. It can be seen that the best attribute subsets selected by the FINEN, NEIEN, and ARNRJE algorithms are the same as each other in some situations, and the number of attributes selected by ARNRJE is less than those of FINEN and NEIEN in the majority of cases. For the Wine, Sonar, and Segmentation data sets, the numbers of attributes selected by the three reduction algorithms are different, where the ARNRJE exhibits the best performance. The slight differences of the three data sets may be caused by the fact that the selected attribute subsets are obtained through reducing the whole data set. For the Wdbc, Wpbc, Prostate, DLBCL, Leukemia and Tumors data sets, the selected attribute subsets are different in general, but the numbers of attributes selected by the three algorithms are very close to each other. Therefore, the proposed ARNRJE algorithm is efficient in dimension reduction of low-dimensional and high-dimensional data sets.

### 4.4. Comparison of Classification Results with Six Reduction Methods on Two Different Classifiers

To further demonstrate the classification performance of our proposed method, six methods are used to evaluate the classification accuracy on the selected attributes. The ARNRJE algorithm is compared with the five related reduction methods, which include: (1) the original data processing method (ODP), (2) the neighborhood rough set algorithm (NRS) [22], (3) the fuzzy rough dependency constructed by intersection operations of fuzzy similarity relations algorithm (FRSINT) [53], (4) the FINEN algorithm [52,58], and (5) the NEIEN algorithm [18]. The two classifiers (3NN and LibSVM) in WEKA are employed to test the classification accuracy. Table 4 denotes the average sizes of attribute subsets selected by the six methods using 10-fold cross validation. What’s more, the corresponding classification accuracy of selected attributes under the 3NN and LibSVM classifiers with 10-fold cross validation is shown in Table 5 and Table 6, respectively.

From Table 4, comparing the average sizes of selected attribute subsets by using 10-fold cross validation, the FRSINT, NEIEN, and ARNRJE algorithms are obviously superior to the RS and FINEN algorithms, but the ARNRJE algorithm is slightly inferior to the FRSINT and NEIEN algorithms. From Table 5 and Table 6, the difference among the six methods can be clearly identified. Then, it can be clearly observed that the classification accuracy of the proposed ARNRJE algorithm outperforms that of the other five methods on most of the nine data sets, except for the Segmentation, Wdbc, and Tumors data sets under the 3NN classifier and the Sonar, Segmentation, Wdbc, and Tumors data sets under the LibSVM classifier. Furthermore, the average classification accuracy of the ARNRJE is the highest on the LibSVM classifier, and has greatly improvement, but the ARNRJE is 0.2% lower than that of FINEN in classification accuracy on the 3NN classifier. From Table 4 and Table 5 under 3NN classifier, although ARNRJE is not as well as FRSINT and NEIEN in the average sizes of selected attribute subsets, the classification accuracy of ARNRJE is nearly 2%–6% higher than that of FRSINT, and that of ARNRJE is approximately 1%–6% higher than that of NEIEN, except for the Segmentation data set. In addition, the classification performance of ARNRJE is better than that of NRS and FINEN on the whole. Though there is some difference in the number of attributes selected by ARNRJE, NRS and FINEN, the accuracy of ARNRJE is higher than that of NRS and FINEN, except for the Wdbc, Segmentation, and Tumors data sets, respectively. The reason is that some important information attributes of the Wdbc and Segmentation data sets are lost in the process of reduction for ARNRJE. Similarly, as seen from Table 4 and Table 6, under the LibSVM classifier, the classification accuracy of ARNRJE is 1%–6% higher than that of FRSINT, ARNRJE is 2%–6% higher than that of NEIEN in classification accuracy, and compared with FINEN, the accuracy of ARNRJE is 0.5%–5% higher, except for the Sonar, Segmentation, Wdbc, and Tumors data sets. For the Wdbc data set, the classification accuracy of ARNRJE is 4% lower than NRS, but ARNRJE selects the less attributes than NRS, and exhibits the better classification accuracy than that obtained by NRS on the other six data sets. As far as the average classification accuracy is concerned, our ARNRJE algorithm shows great stability on the 3NN and LibSVM classifiers, whereas the classification accuracy of the ODP, NRS, and FINEN algorithms is slightly unstable. Based on the results in Table 4, it can be seen that for the Sonar, Segmentation, Wdbc, and Tumors data sets, the proposed ARNRJE method reduces some important attributes in the process of reduction, resulting in the decrease of classification accuracy of reduction sets with fewer attributes. The above results show that no algorithm is congruously better than the others for different learning tasks and classifiers. Overall speaking, our proposed approach can obviously reduce the redundant data and outperforms the other related attribute reduction methods. The experimental results show that our method is an efficient reduction method for redundant data sets, and can improve the classification accuracy for most of the data sets. 

In the above experiments, the coarse ordering of the five methods on time complexity is as follows: *O*(FINEN) = *O*(FRSINT) > *O*(NRS) > *O*(NEIEN) > *O*(ARNRJE), where *O*(A) represents the time complexity of A algorithm. The time complexity of NEIEN algorithm is *O*(*n*^2^) [18]. For the UCI data set with low-dimension, the NEIEN algorithm has the lower time complexity. As we know, the number of samples is usually much greater than that of attributes on the UCI data sets in most cases, whereas for the gene expression data sets, the number of genes is much larger than that of samples. Since the time complexity of ARNRJE algorithm is *O*(*mn*), it is less than that of the NEIEN for large-scale and high-dimensional data sets. Although the time complexity of NEIEN is lower than that of ARNRJE on UCI data sets, the classification performance of ARNRJE algorithm is higher than that of NEIEN algorithm in most instances. For the NRS algorithm, the time complexity is *O*(*m*^2^*n*log*n*) [22]. Since the time of the FRSINT algorithm is mainly spent on getting the fuzzy-rough membership of each sample for different decision classes, the FRSINT algorithm runs slowly and its time complexity is *O*(*m^2^n*^2^) [45,53]. In addition, the time complexity of FINEN is also *O*(*m*^2^*n*^2^) [52], which is time-consuming. The reason is that since the FINEN algorithm is based on similarity relation, they need a lot of time to calculate the similarity relation of attributes [38]. Therefore, it can be easily proven that the ARNRJE algorithm has lower time complexity, can effectively reduce the redundancy, improve the classification accuracy, and optimize the classification process of large-scale complex data.

### 4.5. Comparison of Recall Rate with Three Reduction Methods on Two Different Classifiers

The final portion of our experiments is to measure the recall classification index to evaluate the classification performances of three reduction methods on two different classifiers. The recall rate [59] as a metric is employed to assess the classification performance, which is described as:(28)$$R=\frac{TP}{TP+FN},$$
where True Positive (*TP*) denotes the number of positive instances diagnosed correctly, and False Negative (*FN*) represents the number of positive instances detected as negative.

Table 7 and Table 8 demonstrate the testing results of the recall rate with the FINEN, NEIEN with ARNRJE on the nine data sets on the 3NN and LibSVM classifiers, respectively. From Table 7 and Table 8, the ARNRJE algorithm achieves the highest average recall rate under the two classifiers, and outperforms the FINEN and NEIEN algorithms on most of all the nine data sets. It can be seen from Table 7 under the 3NN classifier that ARNRJE is nearly 4% lower than FINEN for Prostate, and 3% for Tumors in recall rate, respectively. From Table 8 under the LibSVM classifier, ARNRJE is slightly inferior to FINEN for Wdbc and NEIEN for Tumors. The reason is that some important information attributes are lost in the process of preliminary dimensionality reduction or reduction for ARNRJE for the Prostate, DLBCL, Wdbc and Tumors data sets. Thus, this causes misclassification of conditional attributes, and leads to the slightly lower recall rate. The above results manifest that for different learning tasks and classifiers, no algorithm can consistently superior to other algorithms. In general, our proposed ARNRJE algorithm has a relatively good classification performance by measuring the recall rate.

## 5. Conclusions

Attribute reduction, one of the important steps in classification learning, can improve the classification performance in most of cases and decrease the cost of classification. Uncertainty measures for calculating distinguishing ability of attribute subsets play an important role in the process of attribute reduction. The neighborhood rough model can effectively solve the reduction problem of numerical and continuous-valued information system. In this paper, a neighborhood rough sets-based attribute reduction method using Lebesgue and entropy measures is proposed to improve the classification performance of continuous data set. Based on Lebesgue and entropy measures, some neighborhood entropy-based uncertainty measures in neighborhood decision systems is investigated. Then, the neighborhood roughness joint entropy is presented for handling the uncertainty and noisy of neighborhood decision systems, which combines the algebraic view and the information view in neighborhood rough sets. Moreover, their corresponding properties and relationships are discussed. Thus, a heuristic search algorithm is constructed to improve the computational efficiency of selected attributes in neighborhood decision systems. The experimental results show that our proposed algorithm can obtain a small, effective attribute subset with great classification performance.

## Figures and Tables

**Figure 1 entropy-21-00138-f001:**
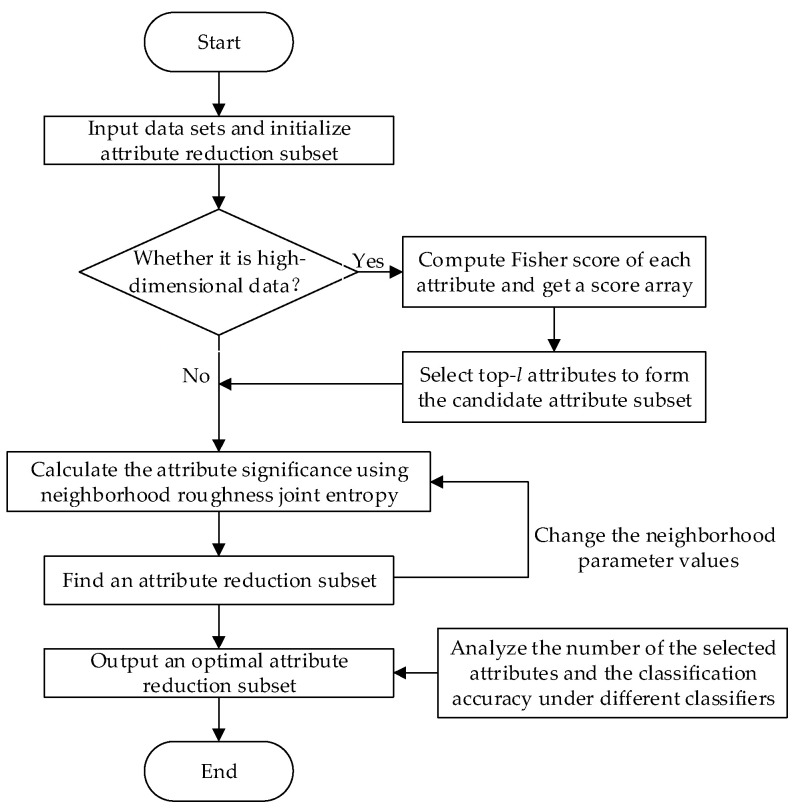
Flowchart of the attribute reduction algorithm for data classification.

**Figure 2 entropy-21-00138-f002:**
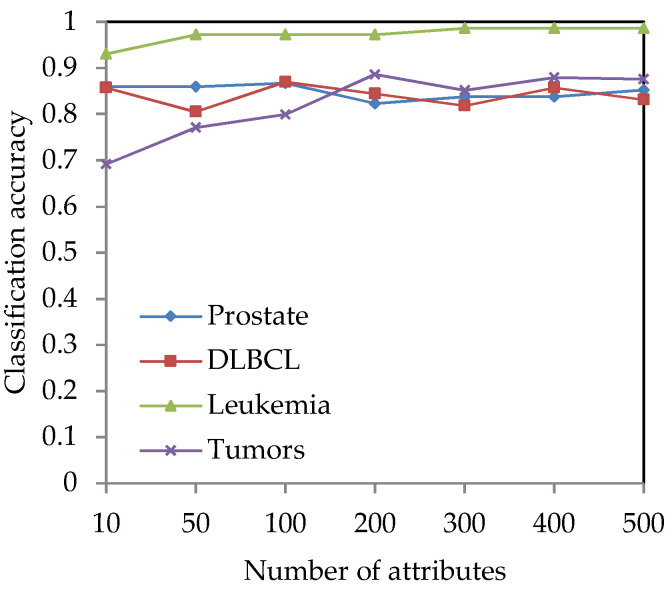
The classification accuracy versus the number of genes on the four gene expression data sets.

**Figure 3 entropy-21-00138-f003:**
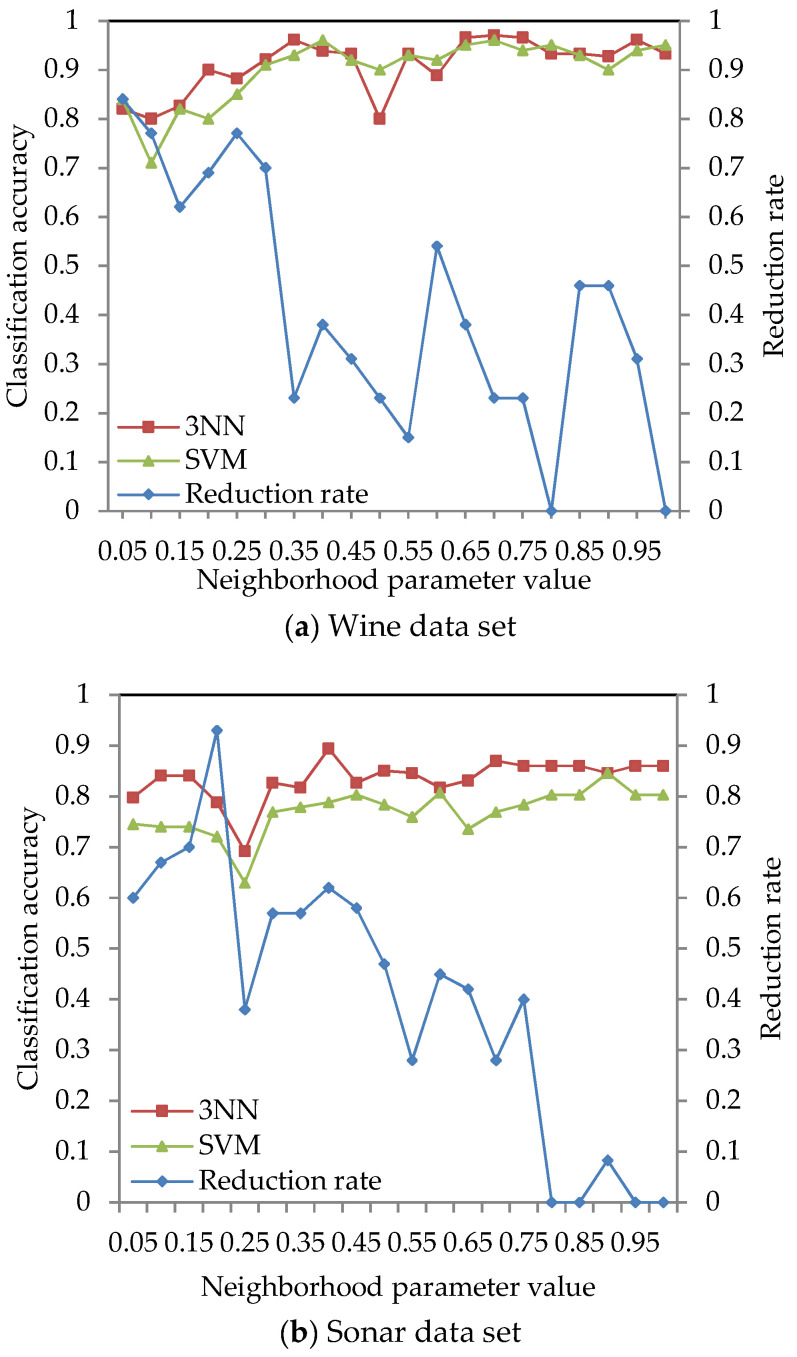
Reduction rate and classification accuracy for nine data sets with neighborhood parameter values.

**Table 1 entropy-21-00138-t001:** A neighborhood decision system.

*U*	*a*	*b*	*c*	*d*
*x* _1_	0.12	0.41	0.61	*Y*
*x* _2_	0.21	0.15	0.14	*Y*
*x* _3_	0.31	0.11	0.26	*N*
*x* _4_	0.61	0.13	0.23	*N*

**Table 2 entropy-21-00138-t002:** Description of the seven public data sets.

No.	Data sets	Samples	Attributes	Classes	Author
1	Wine	178	13	3	Faris et al. [56]
2	Sonar	208	60	2	Wang and Li [57]
3	Segmentation	2310	19	7	Liu et al. [38]
4	Wdbc	569	31	2	Li et al. [39]
5	Wpbc	198	33	2	Chen et al. [13]
6	Prostate	136	12600	2	Mu et al. [24]
7	DLBCL	77	5469	2	Sun et al. [8]
8	Leukemia	72	11225	3	Sun et al. [23]
9	Tumors	327	12558	7	Wang et al. [45]

**Table 3 entropy-21-00138-t003:** The reduction results and the number of selected attributes with the three reduction algorithms.

Data Sets	FINEN	NEIEN	ARNRJE	*δ*
Wine	{1, 2, 3, 4, 6, 7, 8, 9, 10 11, 12, 13}/12	{1, 2, 3, 4, 5, 7, 10, 11, 12, 13}/10	{1, 2, 3, 5, 7, 10, 11, 13}/8	0.65
Sonar	{1, 5, 9, 10, 11, 12, 18, 19, 22, 26, 27, 28, 29, 32, 35, 36, 37, 40, 45, 46, 48, 53, 57, 58, 59, 60}/26	{6, 10, 11, 12, 15, 17, 18, 20, 21, 23, 24, 26, 28, 29, 30, 32, 33, 36, 37, 39, 40, 41, 42, 45, 48, 50, 54, 57}/29	{2, 3, 4, 5, 9, 10, 11, 12, 13, 14, 16, 22, 24, 30, 32, 34, 36, 37, 38, 39, 46, 57, 60}/23	0.4
Segmentation	{2, 5, 6, 7, 11, 12, 13, 17, 18}/9	{2, 5, 6, 7, 11, 12, 13, 17, 18}/9	{2, 5, 6, 10, 11, 13, 17, 18}/8	0.75
Wdbc	{7, 8, 10, 12, 13, 16, 21, 22, 25, 27, 28, 29}/12	{1, 7, 8, 10, 13, 16, 21, 22, 25, 27, 28, 29}/12	{6, 8, 9, 11, 12, 14, 16, 19, 20, 25, 27, 28, 29}/13	0.35
Wpbc	{1, 12, 13, 16, 24, 32}/6	{1, 5, 12, 24, 32}/5	{2, 19, 23, 24, 29, 31}/6	0.55
Prostate	{4483, 6185, 8129, 8623, 8850, 9850, 10753, 12067}/8	{4483, 4847, 6185, 6627, 8623, 8850, 9587, 12067}/8	{11052, 6185, 8986, 5486, 6392, 5757, 8850, 4483}/8	0.9
DLBCL	{453, 1570, 1698, 3127, 3257, 4767}/6	{453, 2930, 3574, 4767, 5283}/5	{453, 1479, 1570, 3127, 3257, 4767}/6	0.5
Leukemia	{2833, 6720, 5555, 10127, 10038, 4839, 8952, 9053}/8	{2833, 6720, 5555, 10127, 10038, 3479, 8964, 515}/8	{461, 1787, 1834, 1962, 2131, 2356, 3821, 5552}/8	0.4
Tumors	{2543, 7648, 3264, 6320, 5411, 6671, 8548, 7781, 10126, 6764, 4178, 4448, 8337, 3043, 4831, 3880}/16	{5411, 6320, 7648, 3264, 3324, 6671, 4300, 6079, 6764, 10126, 8397, 8383, 9046, 7922, 10865, 8687, 2132}/17	{3880, 843, 1730, 3342, 6151, 2960, 3264, 3596, 5624, 4026, 7648, 8383, 8332, 9788, 5412, 8556, 3324, 10126}/18	0.25

**Table 4 entropy-21-00138-t004:** Average sizes of attribute subsets selected by the six methods using 10-fold cross validation.

Data Sets	ODP	NRS	FRSINT	FINEN	NEIEN	ARNRJE
Wine	13	9.1	8.1	12.3	10.2	9.5
Sonar	60	24.8	18.7	25.8	28.9	24.6
Segmentation	19	10.7	8.4	9.5	9.2	8.9
Wdbc	30	17.3	11.9	12.1	11.8	13.3
Wpbc	32	11.6	7.8	6.4	5.3	6.2
Prostate	12600	6.5	8.9	8.4	7.7	8
DLBCL	5469	8.3	8.8	6.1	5.3	7.6
Leukemia	11225	14.7	9.8	8.5	8.2	9.1
Tumors	12558	10.6	9.5	15.8	17.1	18.2
Average	4667.3	12.6	10.2	11.7	11.5	11.7

**Table 5 entropy-21-00138-t005:** Classification accuracy of the six methods under the 3NN classifier.

Data Sets	ODP	NRS	FRSINT	FINEN	NEIEN	ARNRJE
Wine	0.9192	0.9453	0.9281	0.9577	0.9620	0.9775
Sonar	0.8605	0.8588	0.8504	0.8513	0.8326	0.8942
Segmentation	0.8714	0.8021	0.9506	0.9504	0.9488	0.8381
Wdbc	0.9432	0.9553	0.9366	0.9226	0.9385	0.9456
Wpbc	0.6667	0.6752	0.6312	0.6613	0.6263	0.6919
Prostate	0.8235	0.8329	0.8503	0.8689	0.8431	0.8897
DLBCL	0.8831	0.9610	0.9635	0.9635	0.9585	0.9740
Leukemia	0.7339	0.9274	0.8655	0.9246	0.886	0.9306
Tumors	0.7074	0.725	0.7239	0.7781	0.7372	0.7194
Average	0.8232	0.8537	0.8556	0.8754	0.8592	0.8734

**Table 6 entropy-21-00138-t006:** Classification accuracy of the six methods under the LibSVM classifier.

Data Sets	ODP	NRS	FRSINT	FINEN	NEIEN	ARNRJE
Wine	0.9210	0.9213	0.9295	0.9503	0.9210	0.9719
Sonar	0.6587	0.7735	0.7909	0.8168	0.8036	0.7885
Segmentation	0.9048	0.8606	0.9438	0.9317	0.9356	0.9095
Wdbc	0.5167	0.9453	0.9362	0.9230	0.9449	0.9051
Wpbc	0.7374	0.7029	0.7002	0.6875	0.7188	0.7374
Prostate	0.8750	0.8353	0.8527	0.9039	0.8691	0.9118
DLBCL	0.8701	0.9231	0.924	0.9051	0.8758	0.9351
Leukemia	0.4679	0.9165	0.9454	0.9122	0.942	0.9583
Tumors	0.2788	0.7516	0.7308	0.7742	0.7712	0.7627
Average	0.6923	0.8478	0.8615	0.8672	0.8647	0.8756

**Table 7 entropy-21-00138-t007:** The recall rate with the three reduction algorithms under 3NN.

Data Sets	FINEN	NEIEN	ARNRJE
Wine	0.961	0.978	1.000
Sonar	0.910	0.937	0.955
Segmentation	0.833	0.833	0.838
Wdbc	0.951	0.953	0.966
Wpbc	0.815	0.753	0.849
Prostate	0.934	0.882	0.890
DLBCL	1.000	0.974	0.974
Leukemia	0.917	0.958	0.979
Tumors	0.567	0.733	0.867
Average	0.876	0.889	0.924

**Table 8 entropy-21-00138-t008:** The recall rate with the three reduction algorithms under LibSVM.

Data Sets	FINEN	NEIEN	ARNRJE
Wine	0.958	1.000	1.000
Sonar	0.771	0.688	0.874
Segmentation	0.567	0.567	0.667
Wdbc	0.930	0.627	0.924
Wpbc	0.737	0.737	0.768
Prostate	0.566	0.566	0.566
DLBCL	0.753	0.753	1.000
Leukemia	0.389	0.389	0.653
Tumors	0.567	0.691	0.566
Average	0.693	0.669	0.780

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
