# Peer review of "A Neighborhood Rough Sets-Based Attribute Reduction Method Using Lebesgue and Entropy Measures"

_entropy, 2019, doi:10.3390/e21020138_

Round 1

Reviewer 1 Report

A heuristic attribute reduction algorithm based on Lebesgue and entropy neighborhood uncertainty measures for data classification is proposed.

The algorithm 1 should be better structured For example, the instruction FOR at lines 3 and 8a is different from the instruction FOR at line 10 (FOR...DO...END FOR); the GOTO type jump instructions should be avoided; it is convenient to use WHILE or DO WHILE instructions, eliminating conditional jumps.

In section 4.2 athe authors should show also the dipendence of precision and recall indexex from the attribute dimension, highlighing the performance of the proposed algorithm for  massive large-scale data..

In section 4.4 it is not very clear how the performance of the proposed method is better than the other five methods used in the comparative tests. Why the authors measure only the accuracy? It is necessary measure also precision and recall classification indexes to evaluate the classification performances.  In addition,the authors should highlight the performance advantages in relation to the data and attribute dimensions of the datasets used in the tests.

Author Response

Dear,

We are very grateful to you for your valuable comments and suggestions. We have carefully revised the paper in accordance with these comments and suggestions. The added and modified parts are shown in red in the revised manuscript (and changes are marked). The main revisions are as follows.

Comment 1: The algorithm 1 should be better structured For example, the instruction FOR at lines 3 and 8a is different from the instruction FOR at line 10 (FOR...DO...END FOR); the GOTO type jump instructions should be avoided; it is convenient to use WHILE or DO WHILE instructions, eliminating conditional jumps.

Response: Thank you very much for your valuable suggestion.

We have corrected this problem and the algorithm 1 is described as follows:

On Pages 12 and 13:

Algorithm   1. ARNRJE

Input: A neighborhood decision system NDS = <U, C                                                                                   D, δ>, and neighborhood parameter δ.

Output: An optimal   reduction set B.

1.          Initialize B   = Ø.

2.          Calculate   NRH (D, C).

3.          For   i = 1 to |C| do  // Obtain the attribute   core

4.            Calculate Siginner(ai,   C, D).

5.            If Siginner(ai, C, D) > 0, then B = B{ai}.

6.          End   for

7.          Let   R = C –   B.

8.          Do  

9.          For   j =1 to |R|

a)         Calculate   NRH(D, B{aj }).

b)         Select   aj to make it satisfy max{ajR|NRH(D, B{aj})},   and if there are multiple attributes that satisfy the maximum, then the front   should be selected.

End   for

10.     Let B =   B{aj} and R =   R {aj}, and calculate   NRH(D, B).

11.     While NRH(D, B)NRH(D, C)

12.     For k = 1 to |B|   do  // Verify the completeness of the reduction   subset

13.       Select akB.

14.       Compute NRH(D, B −   {ak}).

15.       If NRH(D, B{ak}) NRH(D, B), then let B = B {ak}.

16.     End for

17.     Return an optimal reduction set B.

Comment 2: In section 4.2 the authors should show also the dependence of precision and recall indexes from the attribute dimension, high-lighting the performance of the proposed algorithm for massive large-scale data.

Response: Thank you very much for your valuable suggestion.

  Two high-dimensional data sets have been added, and the precision and recall indexes from the attribute dimension, high-lighting the performance of all the compared algorithms for massive large-scale data have been given as follows:

On Page 15: The selected four gene expression data sets with high-dimensional attributes include the Prostate, DLBCL, Leukemia and Tumors data sets [55]. All the data sets are summarized in Table 2.

On Page 16: Figure 2. The classification accuracy versus the number of genes on the four gene expression data sets.

On Page 17: Figs. 3(h) and 3(i) demonstrate that the reduction rate is decreasing as the parameters increase in most situations, and the neighborhood parameter of Leukemia and Tumors data sets can be set to 0.4 and 0.25, respectively.

On Pages 17-20: Figure 3. Reduction rate and classification accuracy for nine data sets with neighborhood parameter values.

On Pages 20 and 21: Table 3. The reduction results and the number of selected attributes with the three reduction algorithms.

On Pages 21 and 22: Table 4 denotes the average sizes of attribute subsets selected by the six methods using 10-fold cross validation, where the bold font indicates the best result. What’s more, the corresponding classification accuracy of selected attributes under the 3NN and LibSVM classifiers with 10-fold cross validation is shown in Tables 5 and 6, respectively.

On Pages 23 and 24: 4.5. Comparison of Recall Rate with Three Reduction Methods on Two Different Classifiers

Comment 3: In section 4.4 it is not very clear how the performance of the proposed method is better than the other five methods used in the comparative tests. Why the authors measure only the accuracy? It is necessary measure also precision and recall classification indexes to evaluate the classification performances. In addition, the authors should highlight the performance advantages in relation to the data and attribute dimensions of the datasets used in the tests.

Response: Thank you very much for your valuable suggestion.

First, on Pages 15-22, similar to Comment 2, the precision from the attribute dimension, high-lighting the performance of all the compared algorithm for massive large-scale data have been given.

Second, on Pages 23 and 24, the subsection 4.5 is added to compare the recall rate with three reduction methods on two different classifiers as follows:

The final portion of our experiments is to measure the recall classification index to evaluate the classification performances of three reduction methods on two different classifiers. The recall rate [59] as a metric is employed to assess the classification performance, which is described as

,

(28)

where True Positive (TP) denotes the number of positive instances diagnosed correctly, and False Negative (FN) represents the number of positive instances detected as negative.

Table 7. The recall rate with the three reduction algorithms under 3NN.

Data sets

FINEN

NEIEN

ARNRJE

Wine

0.961

0.978

1.000

Sonar

0.910

0.937

0.955

Segmentation

0.833

0.833

0.838

Wdbc

0.951

0.953

0.966

Wpbc

0.815

0.753

0.849

Prostate

0.934

0.882

0.890

DLBCL

1.000

0.974

0.974

Leukemia

0.917

0.958

0.979

Tumors

0.567

0.733

0.867

Average

0.876

0.889

0.924

Tables 7 and 8 demonstrate the testing results of the recall rate with the FINEN, NEIEN with ARNRJE on the nine data sets on the 3NN and LibSVM classifiers, respectively. From Tables 7 and 8, the ARNRJE algorithm achieves the highest average recall rate under the two classifiers, and outperforms the FINEN and NEIEN algorithms on most of all the nine data sets. It can be seen from Table 7 under the 3NN classifier that ARNRJE is nearly 4% lower than FINEN for Prostate, and 3% for Tumors in recall rate, respectively. From Table 8 under the LibSVM classifier, ARNRJE is slightly inferior to FINEN for Wdbc and NEIEN for Tumors. The reason is that some important information attributes are lost in the process of preliminary dimensionality reduction or reduction for ARNRJE for the Prostate, DLBCL, Wdbc and Tumors data sets. Thus, this causes misclassification of conditional attributes, and leads to the slightly lower recall rate. The above results manifest that for different learning tasks and classifiers, no algorithm can consistently superior to other algorithms. In general, our proposed ARNRJE algorithm has a relatively good classification performance by measuring the recall rate.

Table 8. The recall rate with the three reduction algorithms under LibSVM.

Data sets

FINEN

NEIEN

ARNRJE

Wine

0.958

1.000

1.000

Sonar

0.771

0.688

0.874

Segmentation

0.567

0.567

0.667

Wdbc

0.930

0.627

0.924

Wpbc

0.737

0.737

0.768

Prostate

0.566

0.566

0.566

DLBCL

0.753

0.753

1.000

Leukemia

0.389

0.389

0.653

Tumors

0.567

0.691

0.566

Average

0.693

0.669

0.780

Thank you once again for your constructive and valuable comments.

Best wishes,

Prof. Jiucheng Xu, Ph.D.,

College of Computer and Information Engineering, Henan Normal University

Email: jiuchxu@gmail.com

Reviewer 2 Report

The paper seem interesting. - However the comparaison of the proposed approach is only based on numerical results with existing benchmarks.

- It will be more rigorous if the authors are able to provide a theoretical comparison with similar World.

- Particularly, it is required to clearly explain the improvement made on the information and algebric viewpoints.

- The authors must improve the literature review for a more sound comparison of their approach using Lebesgue and entropy measures combining algebra view with information view.The following reference may be relevant:

Marco Di Marzio, Stefania Fensore, Agnese Panzera, Charles C. Taylor. Local binary regression with spherical predictors. Statistics & Probability Letters, Volume 144, January 2019, Pages 30-36

Rashed Khanjani Shiraz, Hirofumi Fukuyama, Madjid Tavana, Debora Di Caprio. An integrated data envelopment analysis and free disposal hull framework for costefficiency measurement using rough sets. Applied Soft Computing, Volume 46, September 2016, Pages 204-219

Lenka Halčinová, Ondrej Hutník, Jozef Kiseľák, Jaroslav Šupina. Beyond the scope of super level measures. Fuzzy Sets and Systems, In press, corrected proof, Available online 13 March 2018. https://doi.org/10.1016/j.fss.2018.03.007

Zhifeng Zhang, Janet David. An entropybased approach for assessing the operation of production logistics. Expert Systems with Applications, Volume 119, 1 April 2019, Pages 118-127

Se Rim Park, Soheil Kolouri, Shinjini Kundu, Gustavo K. Rohde. The cumulative distribution transform and linear pattern classification. Applied and Computational Harmonic Analysis, Volume 45, Issue 3, November 2018, Pages 616-641

Author Response

Dear,

We are very grateful to you for your valuable comments and suggestions. We have carefully revised the paper in accordance with these comments and suggestions. The added and modified parts are shown in red in the revised manuscript (and changes are marked). The main revisions are as follows.

Comment 1: However the comparison of the proposed approach is only based on numerical results with existing benchmarks. It will be more rigorous if the authors are able to provide a theoretical comparison with similar World.

Response: Thank you very much for your valuable suggestion.

We have added a theoretical comparison in the subsection 3.3 as follows:

On Pages 11 and 12:

3.3. Comparison analysis with two representative reducts

It is known that the definition of reducts from the algebra view is usually equivalent to its definition from the information view in a general information system. What’s more, the relative reduct of a decision system in the information view includes that in the algebra view. Thus, Wang et al. [46] declared that any relative reduct of a decision system in the information view must be its relative reduct in the algebra view, so that some heuristic algorithms can be designed further using this conclusion. Based on the ideas of the classification in [46], the definition of reducts based on neighborhood roughness joint entropy of a neighborhood decision system should be developed from the algebra view and the information view in neighborhood rough set theory. For convenience, the reduct in Definition 13 is named as the neighborhood entropy reduct. Liu et al. [38] presented a reduct based on positive region in the neighborhood decision system similar with classical rough set model. This representative relative reduct based on positive region is called the algebra view of the neighborhood rough set theory. Chen et al. [20] defined information quantity similar to information entropy to evaluate the neighborhood classes, used the joint entropy gain to evaluate the significance of a selecting attribute, and proposed a representative joint entropy gain-based reduction algorithm, which is called a reduct in the information view of neighborhood rough sets.

Given a neighborhood decision system NDS = <U, C                                               D, δ> with non-empty infinite set U, BC and D = {d}. Then, a positive region reduct of the neighborhood decision system is presented as follows in [38]: for any aB, if |POSB(D)| = |POSC(D)| and |POSB{a}(D)| < |POSB(D)|, where POSB(D) =  is the positive region of D with respect to B, B is a relative reduct of the neighborhood decision system.

Proposition 7. Given a neighborhood decision system NDS = <U, CD, δ> with non-empty infinite set U, and BC, if B is a neighborhood entropy reduct of the neighborhood decision system, then B is a positive region reduct of the neighborhood decision system.

Proof. Let U = {x1, x2, …, xn, …}, and U/D = {d1, d2, …, dt, …}. Suppose that for a subset BC, it follows from Definition 13 that if NRH(D, B) = NRH(D, C), and for any aB, there exists NRH(D, B) > NRH(D, B{a}), then B is a neighborhood entropy reduct of C relative to D. When NRH(D, B) = NRH(D, C), it can be obtained from Proposition 6 that , and hold, where any xU and 1 ≤ j. By Eq. (13), one has that . So it is obvious that POSB(D) = POSC(D), i.e., |POSB(D)| = |POSC(D)|. For any aB, B{a}B, and from Theorem 1 in [38], one has that , so that POSB{a}(D)POSB(D) holds. Because for any aB, there exists NRH(D, B) > NRH(D, B{a}), thus  holds. It follows that POSB{a}(D)POSB(D). Thus, |POSB{a}(D)| < |POSB(D)| for any aB. Therefore, B is a positive region reduct of the neighborhood decision system.

Notably, the inverse relation of this proposition generally does not hold. According to the above discussions, Proposition 7 shows that the definition of the neighborhood entropy reduct includes that of positive region reduct in the algebra view.

Given a neighborhood decision system NDS = <U, CD, δ> with non-empty infinite set U, BC and D = {d}. For any aB, a reduct of the neighborhood decision system, named as the entropy gain reduct is proposed in [20] as follows: if H(Bd) = H(Cd) and H({B{a}}d) < H(Bd), where  describes the joint entropy of B and d, B is an entropy gain reduct of the neighborhood decision system.

Proposition 8. Given a neighborhood decision system NDS = <U, CD, δ> with non-empty infinite set U and BC, then B is a neighborhood entropy reduct of the neighborhood decision system if and only if B is an entropy gain reduct of the neighborhood decision system.

Proof.  Let U = {x1, x2, …, xn, …}, and U/D = {d1, d2, …, dt, …}. Suppose that for a subset BC, it follows from Definition 13 that if NRH(D, B) = NRH(D, C), and for any aB, there exists NRH(D, B) > NRH(D, B{a}), then B is a neighborhood entropy reduct of C relative to D. Similar to the proof of Proposition 7, when NRH(D, B) = NRH(D, C), from Proposition 6, one has that , where any xU and 1 ≤ j. It is obvious that H(Bd) = H(Cd). Since B{a}B, from Proposition 2 in [20], one has that H({B{a}}d) ≤ H(Bd). Because for any aB, there exists NRH(D, B) > NRH(D, B{a}), so H({B{a}}d) < H(Bd) holds. Hence, B is an entropy gain reduct of the neighborhood decision system.

 Suppose that for a subset BC, and any aB, if H(Bd) = H(Cd) and H({B{a}}d) < H(Bd), then B is an entropy gain reduct of C relative to D. Similar to the proof of Proposition 6, when, by Eqs. (16), (17) and (19), one has that , and then it is obvious that , where any xU and 1 ≤ j. Thus, it can be obtained from Eq. (24) that NRH(D, B) = NRH(D, C). Because B{a}B, it follows from Proposition 6 that NRH(D, B{a})NRH(D, B). Since for any aB, there exists H({B{a}}d) < H(Bd). So, one has that NRH(D, B{a}) < NRH(D, B). Therefore, B is a neighborhood entropy reduct of the neighborhood decision system.

Proposition 8 shows that in a neighborhood decision system, the neighborhood entropy reduct is equivalent to the entropy gain reduct in the information view. According to Propositions 7 and 8, it can be concluded that the definition of neighborhood entropy reduct includes two representative reducts proposed in the algebra view and the information view. Therefore, the definition of neighborhood entropy reduct denotes a mathematical quantitative measure to evaluate the knowledge uncertainty of different attribute sets in neighborhood decision systems.

Comment 2: Particularly, it is required to clearly explain the improvement made on the information and algebric viewpoints.

Response: Thank you very much for your valuable suggestion.

We have given this explanation of the improvement on the information and algebraic viewpoints as follows:

On Page 3: As we can see, many existing methods of attribute reduction in neighborhood rough sets usually only start from the algebraic point of view or the information point of view, while the definition of attribute significance based on algebraic view only describes the effect of attributes on the subset of classification contained [46].

Although these methods have their own advantages, they are still inefficient and not suitable for reducing large-scale high-dimensional data, and the enhanced algorithms only decrease the computation time to a certain extent [47]. Inspired by this, to study neighborhood rough sets from the two views and achieve great uncertainty measures in neighborhood decision systems, the algebra view and the information view will be combined to develop attribute reduction algorithm for infinite sets in continuous-valued data sets.

On Page 10: It is noted that Wang et al. [46] stated that all conceptions and computations in rough set theory based on the upper and lower approximation sets are called the algebra view of the rough set theory, and the notions of information entropy and its extensions are called the information view of rough sets. It follows from Eq. (24) that  is the neighborhood roughness of di with respect to B in the algebra view and it represents the degree of incompleteness of obtaining knowledge of set di, and  is the definition of joint entropy in the information view. Hence, Definition 8 can efficiently analyze and measure the uncertainty of neighborhood decision systems based on Lebesgue and entropy measures from both the algebra view and the information view.

Comment 3: The authors must improve the literature review for a more sound comparison of their approach using Lebesgue and entropy measures combining algebra view with information view. The following reference may be relevant:

Response: Thank you very much for your valuable suggestion.

We have improved the literature review and added the assigned relevant references as follows:

First, on Page 2: Halcinová et al. [28] investigated the weighted Lebesgue integral by Lebesgue differentiation theorem, and used Lebesgue measure to develop the standard weighted Lp-based sizes. Park et al. [29] expressed a measurable map through Lebesgue integration to define the Cumulative Distribution Transform of density functions. Recently, scholars [30-32] introduced Lebesgue measure as a promising additional method to resolve some problems in the application of data analysis.

On Page 26:

Halčinová, L.; Hutník, O.; Kiseľák, J.; Šupina, J. Beyond the scope of super level measures. Fuzzy Sets and Systems 2018, DOI. https://doi.org/10.1016/j.fss.2018.03.007.

Park, S.R.; Kolouri, S.; Kundu, S.; Rohde, G.K. The cumulative distribution transform and linear pattern classification. Applied and Computational Harmonic Analysis 2018, vol. 45, no. 3, pp. 616–641.

Marzio, M.D.; Fensore, S.; Panzera, A.; Taylor, C.C. Local binary regression with spherical predictors. Statistics & Probability Letters 2019, vol. 144, pp. 3036.

Khanjani Shiraz, R.K.; Fukuyama, H.; Tavana, M.; Di Caprio, D. An integrated data envelopment analysis and free disposal hull framework for cost-efficiency measurement using rough sets. Applied Soft Computing 2016, vol. 46, pp. 204–219.

Zhang, Z.F.; David, J. An entropy-based approach for assessing the operation of production logistics. Expert Systems With Applications 2019, vol. 119, pp. 118–127.

Second, to the best of my knowledge, Halcinová et al. [28] used Lebesgue measure to develop the standard weighted Lp-based sizes. Park et al. [29] expressed a measurable map through Lebesgue integration to define the Cumulative Distribution Transform of density functions. Marzio et al. [30] only employed the Lebesgue measure as parametrization of a point to denote the tangent-normal decomposition in the unit hypersphere. Shiraz et al. [31] introduced Lebesgue measure to only study the upper trust and the lower trust in rough space. Zhang et al. [32] used Borel measurable function as the hypothesis to guarantee the existence of Lebesgue–Stieltjes integral.

In our manuscripts, since the Lebesgue measure can efficiently evaluate infinite sets, it is therefore necessary to employ Lebesgue measure to study uncertainty measures and efficient reduction algorithms for infinite sets in neighborhood decision systems. Then, the Lebesgue measure [25] is introduced into neighborhood entropy to investigate the uncertainty measures in neighborhood decision systems, an attribute reduction method using Lebesgue and entropy measures is presented, and then a heuristic search algorithm is designed to analyze the uncertainty and noisy of continuous and complex data sets.

In general, since our motivation is different from those of [28-32], the comparison with the other approaches using Lebesgue measure in [28-32] is very difficult to be obtained.

Thank you once again for your constructive and valuable comments.

Best wishes,

Prof. Jiucheng Xu, Ph.D.,

College of Computer and Information Engineering, Henan Normal University

Email: jiuchxu@gmail.com

Reviewer 3 Report

 Accept in present form

Author Response

Dear,

We are very grateful to you for your valuable comments and suggestions.

Best wishes,

Prof. Jiucheng Xu, Ph.D.,

College of Computer and Information Engineering, Henan Normal University

Email: jiuchxu@gmail.com

Round 2

Reviewer 1 Report

In this new version of their manuscript the authors take into account all my suggestions. I consider this paper publishable in the present form.